Corrected: Author correction

# Microbially induced potassium enrichment in Paleoproterozoic shales and implications for reverse weathering on early Earth

Jérémie Aubineau[1], Abderrazak El Albani[1], Andrey Bekker[2], Andrea Somogyi [3], Olabode M. Bankole[1], Roberto Macchiarelli[4,5], Alain Meunier[1], Armelle Riboulleau[6], Jean-Yves Reynaud[6] & Kurt O. Konhauser[7]

Illitisation requires potassium incorporation into a smectite precursor, a process akin to reverse weathering. However, it remains unclear whether microbes facilitate $K^+$ uptake to the sediments and whether illitisation was important in the geological past. The 2.1 billion-year-old Francevillian Series of Gabon has been shown to host mat-related structures (MRS) and, in this regard, these rocks offer a unique opportunity to test whether ancient microbes induced illitisation. Here, we show high K content confined to illite particles that are abundant in the facies bearing MRS, but not in the host sandstone and black shale. This observation suggests that microbial biofilms trapped $K^+$ from the seawater and released it into the pore-waters during respiration, resulting in illitisation. The K-rich illite developed exclusively in the fossilized MRS thus provides a new biosignature for metasediments derived from K-feldspar-depleted rocks that were abundant crustal components on ancient Earth.

[1] UMR 7285 CNRS IC2MP, University of Poitiers, Poitiers 86073, France. [2] Department of Earth and Planetary Sciences, University of California, Riverside, CA 92521, USA. [3] Nanoscopium Beamline Synchrotron Soleil, BP 48, Saint-Aubin, Gif-sur-Yvette 91192, France. [4] Department of Geosciences, University of Poitiers, Poitiers 86073, France. [5] Department of Prehistory, UMR 7194 CNRS, National Museum of Natural History, Paris 75005, France. [6] UMR 8187 CNRS LOG, University of Lille, ULCO, Villeneuve d'Ascq 59655, France. [7] Department of Earth and Atmospheric Sciences, University of Alberta, Edmonton, AB T6G 2E3, Canada. Correspondence and requests for materials should be addressed to A.E.A. (email: abder.albani@univ-poitiers.fr)

The conversion of smectite to illite–smectite mixed-layer minerals (I–S MLMs) is a multistep process that takes place during diagenesis, and is mainly controlled by temperature, time, and $K^+$ availability[1–4]. The illitisation process progressively converts hydrated smectite-rich layers to illite-rich layers, resulting in a common co-occurrence of smectite-rich, randomly interstratified I–S MLMs and long-range ordered I–S MLMs[5,6]. Two main mechanisms for smectite to illite transformation have been inferred, and these are likely linked to specific environmental conditions that are yet to be defined[3,5–8]. A solid-state transformation characterized by a progressive replacement of smectite with illite via tetrahedral/octahedral Al substitutions and interlayer exchange has been proposed[3,7]. The most likely mechanism involves the dissolution of the smectite lattice and growth of illite crystals due to the Ostwald ripening-like effect[5,6,8].

Microorganisms are capable of precipitating and transforming clay minerals as a result of their high surface reactivity and metabolic activity[9–17]. Perhaps one of the most striking examples is the transformation of hydrated smectite to illite through a process that involves the microbial release of structural Fe(III) from the smectite lattice via dissimilatory iron reduction (DIR)[13–15]. This results in the negative charge of the octahedral sheet and $K^+$ uptake into the interlayer spaces to balance the structural charge imposed by DIR, thus enhancing the illitisation process. In laboratory experiments, this clay transformation is enhanced by an abundant supply of $K^+$ cations to the medium[13,15]. In natural systems, K-rich fluids and dissolution of K-feldspar may also provide $K^+$ for this reaction to take place[17]. A biologically controlled $K^+$ addition to sediments during diagenesis has not yet been described.

The Francevillian Formation B (FB) marine sedimentary rocks are exquisitely preserved within the Francevillian basin in southeastern Gabon[18] (Fig. 1a; Supplementary Fig. 1a). The maximum burial depth is estimated to reach about 2 km with minimal burial temperature[19–21]. An assemblage of diagenetic illite, chlorite, I–S MLMs, and K-bentonite characterizes the FB Formation[19,22,23]. These sediments were deposited in an oxygenated environment[24] ~2.1 billion years (Ga) ago[25] during a transgression (FB₁ Member) followed by a sea-level fall (FB₂ Member). The shallowing brought the depositional site to the photic zone and allowed the formation of mat-related structures (MRS) under the dominant influence of cyanobacteria[26,27]. The marine $FB_2$ Member, which bears the MRS, is essentially a massive sandstone deposit (FB2a unit) overlain by black shales and diagenetic carbonates interbedded with thin siltstone layers (FB2b unit; Supplementary Fig. 1b). Here, we present whole-rock and in situ geochemical analyses, bulk XRD, electron microscopy images, and synchrotron-based elemental distribution maps from MRS and their host sediments. We studied samples from a 15-m-thick, coarse-grained sandstone that conformably underlies a 5-m-thick, thinly-bedded black shale that hosts the oldest large colonial macrofossils with evidence for organism motility previously described from the $FB_2$ Member of the FB Formation[28–30] (Supplementary Fig. 1b). In this study, we highlight that K enrichments are localized in illite particles within fossilized MRS, and, most crucially, that this provides insight for the search of biosignatures in early Earth's sediments where stable isotope evidence are controversial. We discuss how these findings have important implications for Earth's climate and ocean chemistry in the Paleoproterozoic.

## Results

**Textures of mat-related structures.** The investigated MRS exhibit a range of surface morphologies including those generated through mat propagation (mat-growth structures) and mat preservation/protection of structures formed independently from mat growth (mat-protected structures)[26,27,31]. The elephant-skin texture (Fig. 1b) is one of the most common mat-growth morphologies showing evidence for biological activity (i.e., undirected gliding motility, phototactic behavior[32,33]), while wrinkle and Kinneyia structures, linear ridges, and nodule-like structures point to the biostabilization of sediment. Mat laminae associated with irregular rounded pits (Fig. 1c) provide evidence for biofilms that were capable of bioweathering the substratum onto which they were attached[34]. Scanning electron

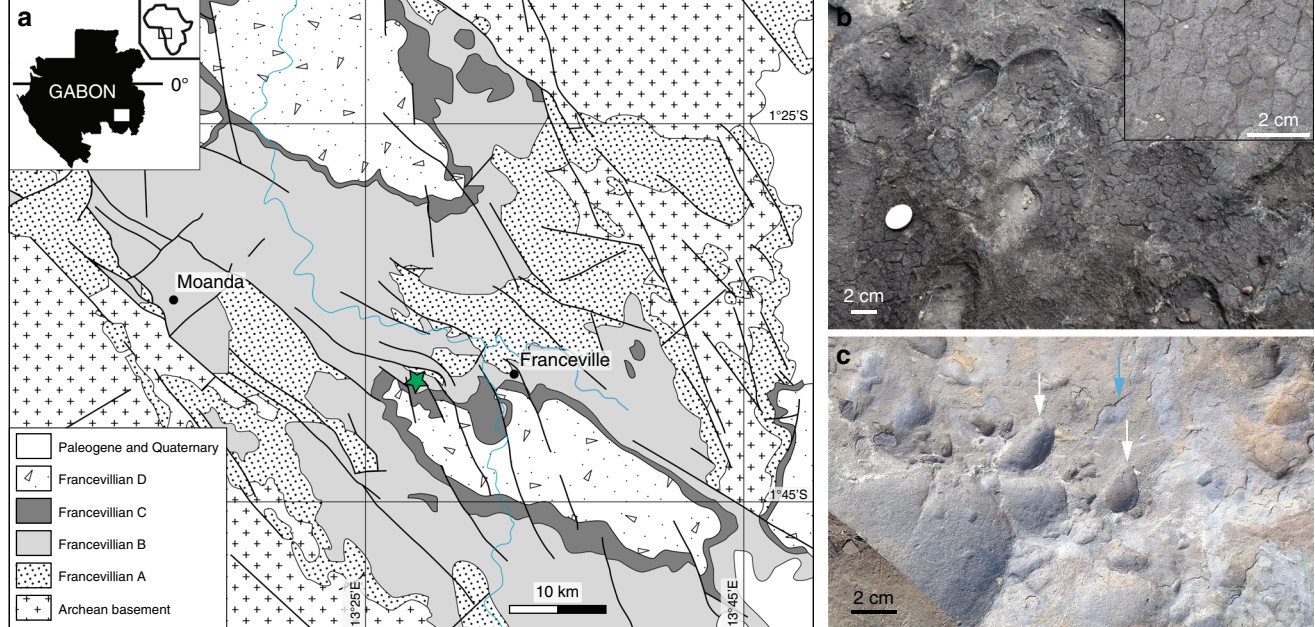

**Fig. 1** Geological map and field photographs of the $FB_2$ Member outcrops, Gabon. **a** Geological map of the Francevillian basin adapted from Bouton et al.[18]. The studied area is the Moulendé Quarry (green star). **b** Elephant-skin texture on the bedding plane of coarse-grained sandstones. Inset box shows reticulate patterns. **c** Micrometer-thick microbial mat laminae (blue arrow) on the bedding plane of sandstones with rounded pits (white arrows)

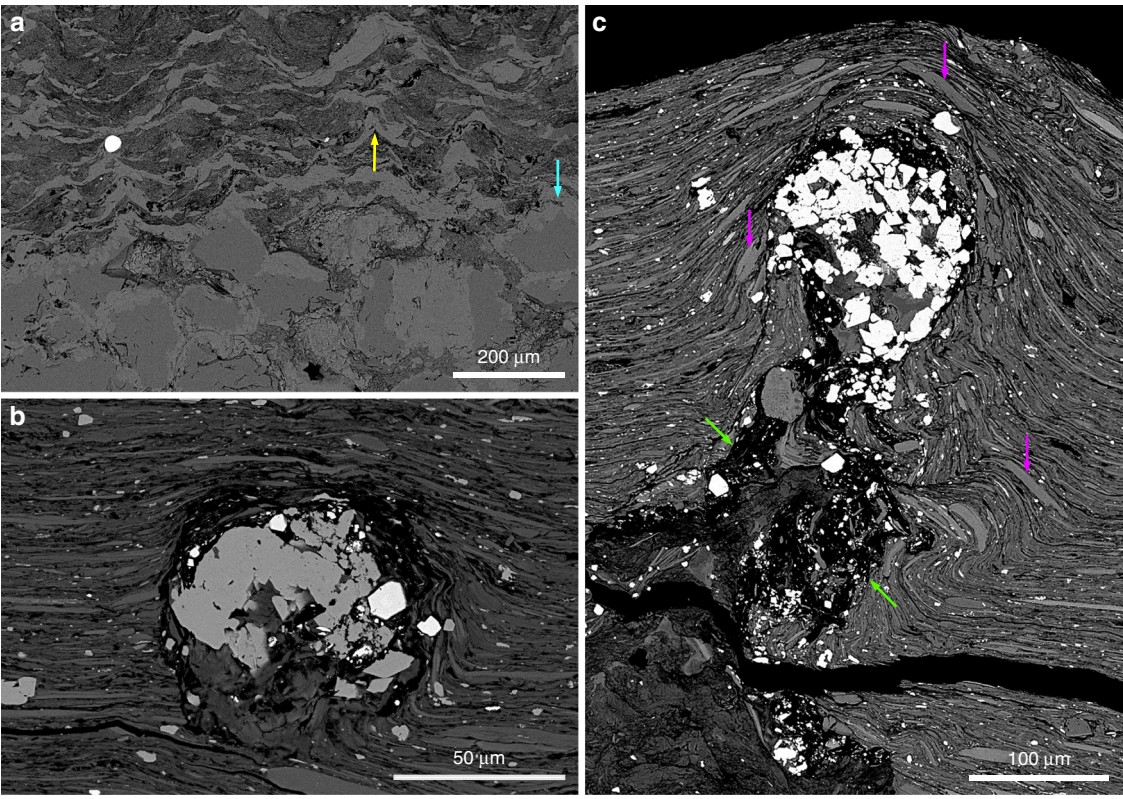

**Fig. 2** Biogenetic fabrics in the mat-related structures. The MRS textures are shown through SEM images. **a** Tufted microbial fabrics developed above the poorly-sorted quartz sandstone. Yellow and blue arrows point to tufts and quartz grains, respectively. **b** A void-filling titanium oxides that may have filled an oxygen bubble produced within the microbial mat. **c** Nearly-circular void filled with titanium oxides at the tip of a cone-like feature (green arrows) Detrital dioctahedral micas (e.g., muscovite) are shown by purple arrows

microscopy–back-scattered electron imaging (SEM–BSE) coupled to energy-dispersive X-ray spectrometry (EDX) displays textures and mineralogical compositions within the mat laminae that are different from host sediments (Fig. 2; Supplementary Figs. 2 and 3)[26].

Laminated black shales host pyritized MRS that are primarily composed of 20–30 μm euhedral pyrite grains (Supplementary Fig. 2c, d). The total sulfur content (higher than 6 wt%) within these MRS is associated with low total organic carbon content (Supplementary Table 1). In addition, laminations between pyrite crystals, reflecting successive mat growth, were recently documented[26]. The non-pyritized MRS (with less than 4 wt% of S) comprise tufted microbial fabrics above coarse-grained sandstones (Fig. 2a) with abundant nearly circular voids within the mat structures filled with titanium oxides, and they are typically enriched in heavy minerals, such as titanium oxides and zircons (Fig. 2b, c; Supplementary Figs. 2e and 3; Supplementary Table 1). Inferred to be gas escape features, the voids show a pre-compactional formation whereby the clay particles are disorganized around the cone-like structures yet aligned above them.

**Potassium enrichment in the mat-related structures**. Whole-rock geochemical analyses of major elements are presented in Supplementary Table 1. Our data show higher $K_2O/SiO_2$ ratios for the MRS compared to the surrounding sediments, thus indicating a potassium-rich source specific to this lithology (Fig. 3). The observed difference in the dataset is statistically significant ($n = 41$; Kruskal–Wallis test: $\chi^2 = 33.29$, df $= 3$, $p$-value $< 10^{-7}$). Pairwise comparisons show that the difference between all pairs of groups is also statistically significant (Supplementary Table 2). The subtle, but significant difference in

ratios between pyritized mats and black shales from the FB2b unit might be related to contamination of some black shale sediments during the extraction of pyritized mat structures (Fig. 3). Binary plots of selected major elements were used to compare the MRS and host sediments (Supplementary Note 1; Supplementary Fig. 4), as well as the likely control of mineral composition and the effect of quartz dilution on these sediments.

Synchrotron-based scanning X-ray fluorescence microscopy (XFM) study was performed on millimeter-sized areas of representative samples. This non-destructive, element-specific imaging technique provides the distribution of S, K, Ca, Ba, Mn, Fe, Ni, Cu, Zn, Ga, Ge, and As with micron-scale spatial resolution and high analytical sensitivity. The distributions of S and K were imaged in pyritized MRS (Fig. 4a), black shale (Fig. 4b), and non-pyritized MRS (Fig. 4c). Sulfur is restricted to pyrite microcrystals (Fig. 4a, c), highlighting the difference between pyritized and non-pyritized MRS. Pyrite crystals are not enriched in K. Potassium is homogeneously distributed throughout the MRS and black shales, without significant micro-scale heterogeneities. The K-distribution maps reveal a higher K content in the pyritized and non-pyritized MRS compared to that in the host sediments.

**Potassium-rich clay assemblage patterns**. Eleven representative samples, consisting of five MRS and six host sediments from the FB₂ Member, were further analyzed using the X-ray diffraction (XRD) technique (Fig. 5a, d; Supplementary Figs. 5–7). The stacking mode ($R$, Reichweite ordering parameter) of I–S MLMs, ranging from R0 (randomly interstratified) to $R \geq 1$ (long-range ordered) MLMs, was calculated (see Methods) to characterize the illite content with a higher $R$ parameter corresponding to a higher

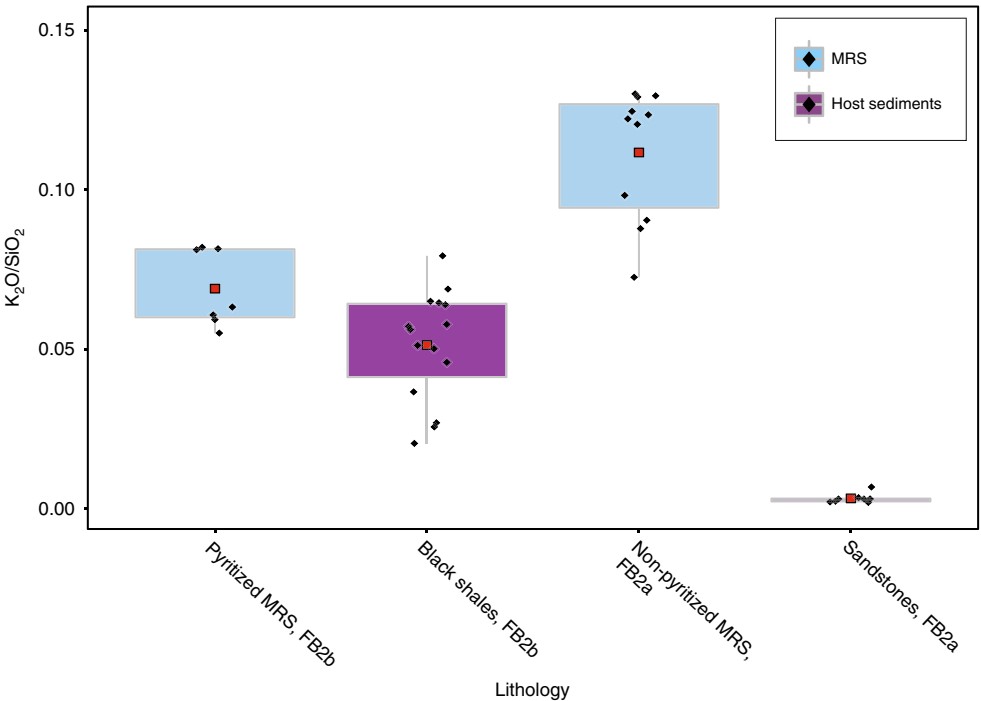

**Fig. 3** $K_2O/SiO_2$ ratios from mat-related structures and host sediments. Pyritized MRS are hosted by black shales and non-pyritized MRS are observed on bedding surface of sandstones. The data for each lithology are represented as box plots with a red square showing the mean, black diamonds corresponding to individual samples, 50% of the data are shown as a box and whiskers extend to 1.5 times the interquartile range

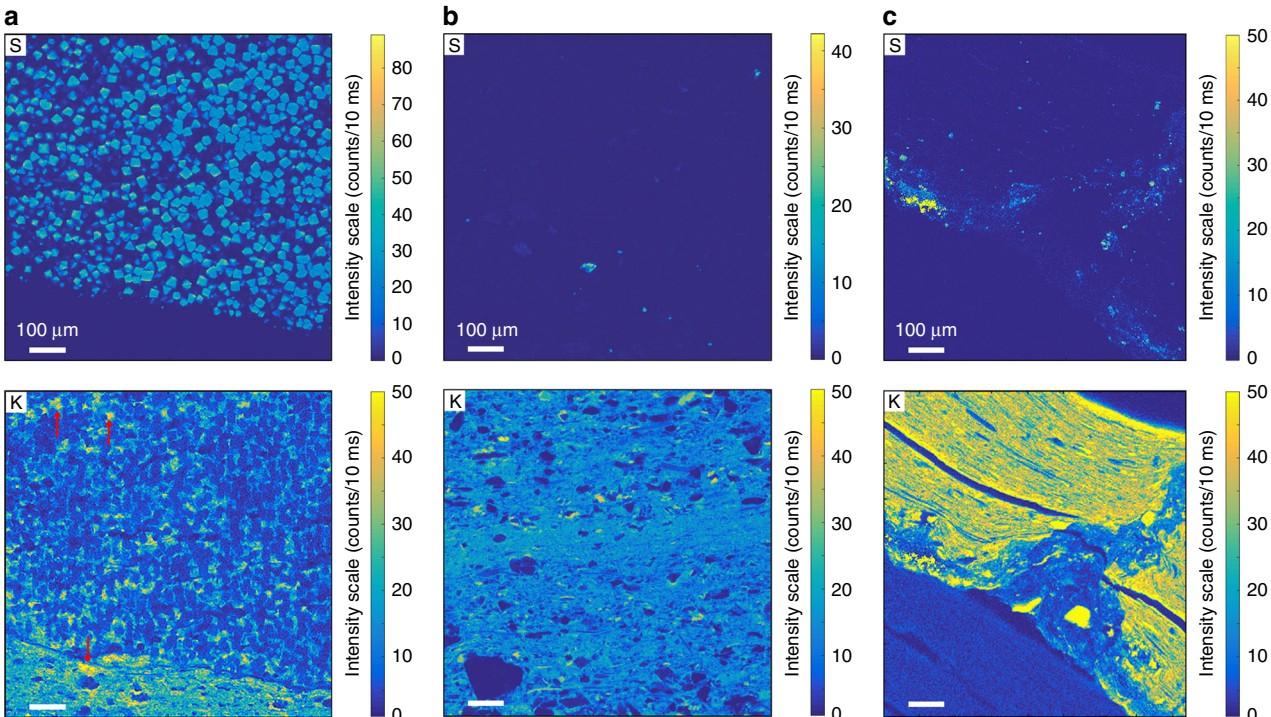

**Fig. 4** High-resolution sulfur and potassium distribution maps. **a** Pyritized MRS. Red arrows point to K enrichment between pyrite crystals and in the host sediment of MRS. **b** Black shales. **c** Non-pyritized MRS. Higher X-ray fluorescence (XRF) intensities correspond to higher S and K contents. For easier comparison, a common intensity scale was chosen for the K distribution maps

illite content. The mineral assemblage of the MRS (both non-pyritized and pyritized types) includes quartz, mica, illite, chlorite, K-rich R3 I–S MLMs, and pyrite. In contrast, the host sediments contain quartz, mica, illite, chlorite, R0 I–S MLMs, and dolomite (Supplementary Figs. 5–7). Notably, K-feldspars have

been shown to be entirely absent, while plagioclase is present in the FB and FC formation lithologies, potentially reflecting provenance composition or depositional rather than climatic (via intense weathering) control[19,35]. XRD patterns of the <2 μm fraction from the MRS show peaks at 10.56–10.37 Å in air-dried

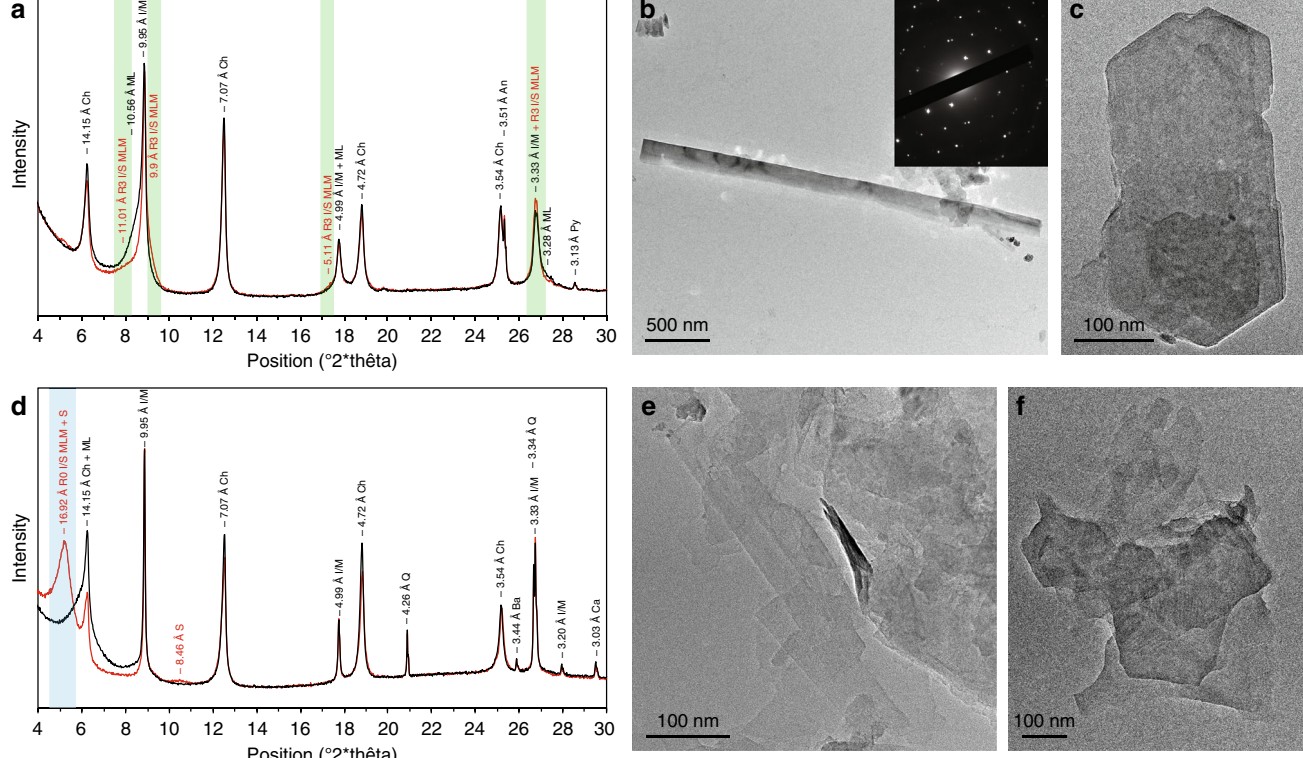

**Fig. 5** Mineralogical composition of the <2 μm clay-size fractions. X-ray diffraction profiles of oriented preparations after air-drying (black lines) and glycolation (red lines) and their transmission-electron images are given. **a** Microbial mat specimen. **b** Large, well-crystallized lath from microbial mat laminae. The inset shows selected area electron diffraction (SAED) pattern. *hk*0 pattern shows a hexagonal structure typical of phyllosilicates and coherently stacked crystals. **c** Hexagonal habit of illite from a mat sample. **d** Associated sandstone sediment. **e** Lathlike and poorly crystallized particles from the host sandstone. **f** Tiny hexagonal-shaped particles from the host sediment. [Green areas correspond to long-range ordered illite–smectite mixed-layer minerals (R3 I/S MLM); Blue areas represent randomly ordered illite–smectite mixed-layer minerals (R0 I/S MLM); smectite (S); chlorite (Ch); mixed-layer (ML); illite/mica (I/M); anatase (An); quartz (Q); pyrite (Py); barite (Ba); calcite (Ca)]

(AD) preparations splitting into 11.08–10.82 Å and ~9.9 Å peaks after ethylene glycolation (EG), consistent with characteristics of R3 I–S MLMs (Fig. 5a; Supplementary Fig. 6a, c). Smectite-rich R0 I–S MLMs, mixed with R3 I–S MLMs, in the host sediments are indicated by the shift of the 001 peak from around 14.35–14.1 Å in AD to 17.22–16.92 Å in EG (Fig. 5d; Supplementary Fig. 6b, d). The 002 peak of pure smectite close to 8.5 Å (Fig. 5d) further characterizes the host sediments.

The modeling results by the NEWMOD program[36] for the MRS confirm the lack of R0 I–S MLMs. In contrast, the relative abundance of R3 I–S MLMs, together with a relative enrichment in detrital mica and chlorite, are higher than in the host sediments (Supplementary Figs. 8 and 9; Supplementary Table 3), suggesting an intense dissolution of the reactive clay fraction and K input. In addition, smectite layers represent less than 12% of the R3 stacking in both the MRS and sediments, but are more abundant in the R0 I–S MLMs (S = 65%).

Transmission electron microscopy (TEM) was used to image the <2 μm clay-size fraction and, specifically, the illite crystal habit. The illites form large laths, up to 10 μm long and 0.6 μm wide (Fig. 5b; Supplementary Fig. 10a–c), as well as hexagonal particles ~0.2–0.5 μm in size in the MRS (Fig. 5c; Supplementary Fig. 10d–f). The large lath-shaped particles indicate a well-crystallized illite growth through an Ostwald ripening-like process in which the small particles are dissolved, while large particles grow at their expense in order to minimize surface free energy[8,37,38]. The hexagonal particles represent steady-state growth during an advanced stage of illitisation. The particle size of illite is much smaller in the host sediments than in the MRS.

Illite in the host sediments predominantly forms lath-shaped particles ~0.15–0.4 μm long and 0.05 μm wide, with a few hexagonal particles up to 0.2 μm in size (Fig. 5e, f; Supplementary Fig. 10g, h). These tiny lath-shaped particles correspond to poorly crystallized illite[38,39].

## Discussion

The textural and geochemical observations of the pyritized MRS suggest that the organic carbon was oxidized via dissimilatory sulfate-reduction (DSR)[29]. This mechanism is also supported by the common diagenetic mineralization of individual mat layers, arguing for the former presence of organic carbon[40]. Moreover, if the earlier interpretation of the MRS in the FB2 Member being cyanobacterial in origin is correct[26], then the tufted appearance might be related to the escape of oxygen bubbles similar to that described in modern and ancient stromatolites[41–43]. The differences between non-pyritized MRS and pyritized MRS might reflect a higher sedimentation rate and energy in the depositional setting of the non-pyritized MRS where a rapid burial would not have allowed sulphidic pore-waters to develop. Alternatively, this difference might be due to less reactive iron initially supplied to the depositional setting where the non-pyritized MRS were formed, consistent with the inferred shallow-water and high energy conditions.

The illitisation process results in different abundances of illite and smectite, but points toward the association of the illite end-member with the MRS. The source of K+ for diagenetic smectite-to-illite transformation is commonly linked to the dissolution of K-rich tuffs, K-feldspars, and micaceous minerals, as well as to

migrating K-rich fluids[39]. Petrographic and XRD observations argue against significant contribution of K-rich tuffs to the composition of the $FB_2$ Member. The survival of R0 I–S MLMs (i.e., smectite-rich) in both the sandstones and black shales lacking the MRS is consistent with incomplete illitisation, attributed to the lack of K-feldspars in the FB Formation sediments[19,35]. In our samples, dioctahedral micas (e.g., muscovite) are only mechanically deformed (Fig. 2c; Supplementary Fig. 3), reflecting a limitation to no chemical alteration under diagenetic conditions[44] and, therefore, they cannot be considered as a significant source of $K^+$. Moreover, it is unlikely that an external abiotic supply (e.g., K-rich fluids or K-bearing minerals) would have provided $K^+$ only to the µm-thick microbial fabrics, but not to the host sediments. Organic matter could delay illitisation by inhibiting the exchange of interlayer cations[45]; nonetheless non-pyritized MRS show both high organic carbon content (Supplementary Table 1) and an advanced stage of illitisation.

Our observations imply instead that microbes played two roles in the illitisation process. First, the large mineral surface area of smectite may have helped to bind organic compounds within interlayer spaces[46]. The concentration of organic matter is critical because illitisation is promoted via DIR, a process that oxidizes the sorbed organic carbon via reduction of Fe(III) minerals, and ultimately releases Fe(II) to the sediment pore-waters[15]. Second, the microbes sequestered $K^+$ into the MRS of the $FB_2$ Member. In the case of the latter, potassium is one of the most essential cations for microorganisms to maintain their cellular machinery[47], and, as a consequence, it is typically enriched in cells relative to seawater[48].

Biologically derived K could have been incorporated into MRS through the $K^+$ uptake into the living microbial cells, and/or $K^+$ absorption by reactive organic macromolecules—containing carboxyl, phosphate, and hydroxyl groups—as part of the cell envelope or extracellular polymeric substances (EPS) of biofilms. In the first process, the structure of the $K^+$ channel (e.g., proteins controlling $K^+$ transport across cell membranes) forms a selectivity filter that prevents $Na^+$ uptake, but works perfectly for incorporating K cations[49]. Thus, $K^+$ contributes to the regulation of osmotic pressure differences, the control of internal pH, and the activation of intracellular enzymes[47]; the thylakoid potassium channel (SynK) even promotes an optimal photosynthesis in cyanobacteria[50]. As a result of K-dependant metabolic reactions, the intracellular concentration of $K^+$ in modern cells (100 mM) is one order of magnitude greater than that in the seawater[48,51], further supporting the ability of microbes to sequester this macronutrient. The second $K^+$ enrichment process involves the electrostatic binding of metal cations from solution to the cell wall and/or EPS, and in turn, the nucleation and mineral precipitation of authigenic mineral phases[52,53]. Indeed, microbial cell surfaces have been shown to concentrate $K^+$ from solution[54], and importantly, increased $K^+$ supply has been shown to stimulate the production of EPS by some bacterial strains[55]. In addition, microbial respiration of EPS during early diagenesis could liberate adsorbed cations into the pore waters, thus inducing $K^+$ supersaturation that can additionally promote mineral authigenesis[56]. Regardless of the path for $K^+$ uptake, biologically derived $K^+$ would have been required to facilitate the smectite-to-illite transformation during both decay of microbial mat organic matter and early diagenesis.

The proposed mechanism of bacterially enhanced illitisation has important implications for the Paleoproterozoic climate and ocean chemistry because the facilitated removal of $K^+$ from seawater into the clay fraction of sediments is akin to the modern reverse weathering process[57] to explain difference in the chemical composition of rivers and seawater. Formation of authigenic clays

in the oceans involves (as an unbalanced reaction): $Si(OH)_4 +$ cations ($K^+$, $Mg^{2+}$, $Fe^{2+}$, $Al^{3+}$) + $HCO_3^- \rightarrow$ clay mineral + $CO_2$ + $H_2O$. In the Amazon continental shelf, reverse weathering was estimated to account for a sink of ~10% of the annual riverine supply of $K^+$ to the oceans[58]. Moreover, the release of $CO_2$ from bicarbonate consumption has been recently inferred to have helped maintain a warm Precambrian Earth at a time of lower solar luminosity and stabilized marine pH at lower values by releasing protons to seawater[59]. While the conversion of smectite to illite is different[60]: smectite + $K^+ \rightarrow$ illite + cations ($Na^+$, $Ca^{2+}$, $Mg^{2+}$, $Fe^{2+}$) + $Si(OH)_4$, its importance lies in the enforcing ultimate removal of $K^+$ from seawater with MRS, which would have strengthened the reverse weathering buffer on global climate during the Precambrian. Notably, in the absence of terrestrial vegetation and limited clay burial on landmasses in the Precambrian to consume $K^+$ released by chemical weathering, terrestrial $K^+$ fluxes to the oceans would have been significantly larger than in the Phanerozoic[61]. The K content in Precambrian seawater is highly uncertain with estimates ranging from those lower than that in the modern seawater to those that are higher[62–65]. Large land masses likely did not emerge until ~2.5 Ga[66], thus the continental margin repository for marine evaporites was smaller with the correspondingly larger content of Na and K in seawater[62,67,68]. Regardless, the ability of bacteria to facilitate the incorporation of $K^+$ into illite at the time when microbial mats were likely more extensive in the oceans in the absence of grazing metazoans[69] provides a previously unrecognized biological feedback on the Earth surface system to control seawater pH and the atmospheric $CO_2$ content.

The recognition of ancient metabolic activity more specific than biogenic graphite is also of paramount importance in the search for Archean biosignatures. In this regard, we suggest that the process of illitisation in association with the MRS might be evidence of biologically supplied $K^+$. Until now, this concept has not been tested because of the common abundance of K-feldspars in Phanerozoic and Precambrian sedimentary basins, which is related to the presence of felsic igneous rocks in the continental crust of all ages[70]. However, the predominance of mafic and ultramafic igneous rocks in the Archean and early Proterozoic greenstone belts[71,72] requires a different mechanism to explain the $K^+$ enrichment with respect to host sediments that lack potassium-rich detrital minerals and illite-rich authigenic clays. Our finding of K-controlled mineral transformation during diagenesis in clay-rich shale deposits provides a new insight for seeking a potential biosignature in the early Earth's sedimentary environments that lacked granitoids in their provenance and where morphologic and carbon isotope evidence for MRS are often compromised.

## Methods

**Sampling and sample preparation.** We collected more than 100 MRS with their associated host sediments from Moulendé Quarry, Francevillian basin, Gabon. The samples were photographed at the University of Poitiers using a Nikon Europe D610 digital single-lens reflex camera equipped with a Nikon AF-S 24–120 mm f/4G ED VR lens. Textural contrast between mat specimens and host sediments was investigated on polished slab sections with a ZEISS Discovery V8 stereoscope coupled with a Axio Cam ERc 5s microscope camera. Thirty samples covering all rock types were selected for mineralogical analyses. The µm-thick mat laminae (both non-pyritized and pyritized, defined according to petrographic analyses[26] and their sulfur content) were carefully removed from their underlying host rocks by using a cutter blade to avoid contamination. Directly below the mat-bearing rocks, sandstone or black shale sediments were gently extracted with a hammer for mineralogical and geochemical analyses. MRS were carefully removed to extract and prepare <2 µm clay fraction and whole-rock powder for mineral and organic carbon analyses.

**X-ray diffraction.** Approximatively the same quantity of the whole-rock powder and clay mineral fraction (<2 µm) from 11 samples were analyzed with a Bruker D8 ADVANCE diffractometer at the University of Poitiers using CuKα radiation

operating at 40 kV and 40 mA. The <2 μm clay fraction was separated by dispersion of gently hand-crushed bulk samples in deionized water with an Elma S60 ultrasonic agitation device without any chemical pre-treatment[73]. The dispersed particles were let to settle under gravity at a controlled room temperature of 20 °C, and centrifuged to separate <2 μm clay fraction. Oriented slides were prepared by drying ~1 mL of suspension on glass slides at room temperature. The <2 μm fraction of some smectite-rich and illite-rich samples has been Ca-saturated after three dispersions in 1 M CaCl₂ solution followed by several washings with deionized water. This cation exchange homogenizes the sheets of swelling clays to keep stable interlayer properties. Analysis of whole-rock powder samples was performed over an angular range of 2–65° 2θ with a step size of 0.025° 2θ per 3 s. Oriented slides of the <2 μm clay fraction were analyzed at a step size of 0.02° 2θ per 3 s counting time and over 2–30° 2θ angular range after successive air drying (AD) and ethylene glycol (EG) saturation. Oriented Ca-saturated samples were then prepared and analyzed in AD and EG states. Background stripping, indexing of peaks, and mineral identification were done using Bruker Eva software by comparing with International Centre for Diffraction Data (ICDD) files. Illite polytypes were observed on randomly oriented powders (<2 μm clay fraction) and analyzed at a step size of 0.01° 2θ per 2 s counting time and over 19–33° 2θ angular range. The results were compared with reference data[74–76]. Illite with 2M₁ and 1Mt polytypes represent detrital and diagenetic illite, respectively[77].

**X-ray diffraction profile modeling**. The stacking mode (R, Reichweite ordering parameter) of I–S MLMs ranging from randomly interstratified (R = 0) MLMs to long-range ordered MLMs (R = 1, 2, and ≥3) as well as the proportion of smectite and illite layers were determined by fitting experimental Ca-saturated samples in AD and EG states (over 2–30° 2θ CuKα) using the NEWMOD 2.0 program simulation[36]. When the same solution is obtained for the AD and EG states, we assumed the fit is acceptable. The instrumental and experimental factors such as horizontal and vertical beam divergences, goniometer radius, length, and thickness of the oriented slides were introduced without further adjustment[78]. Sigmastar (degree of orientation of the clay particles on the slide) was set between 12° and 18° and the mass absorption coefficient (μ*) to 45 cm²/g, as suggested by Moore and Reynolds[73]. Angular domains containing non-clay mineral reflections were not included in the profile-fitting process.

**Electron microscopy**. Polished slab sections of selected samples were carbon coated and imaged with a FEI Quanta 200 scanning electron microscope (SEM) equipped with an energy dispersive X-ray spectrometer (EDX) at the University of Lille. Mineral identification and documentation of textural relationships were acquired in back-scattered electron mode (BSE) operated at accelerating voltage of 15 kV, 1 nA beam current, and a working distance of 10.5 mm. TEM analysis of the <2 μm clay fraction was performed with a JEOL 2100 UHR microscope working at 200 kV accelerating voltage with a LaB6 emitter and equipped with a JEOL EDX detector at the University of Poitiers. Samples were prepared for TEM analysis by suspending ~1 mg of each sample in distilled water. A droplet of diluted suspension after ultrasonic treatment was deposited on a C-coated Au grid. Lath- and hexagon-shaped particles were selected from the grids because they only correspond to illitic minerals[37–39]. Selected-area electron diffraction (SAED) patterns were collected to image the hexagonal net pattern of the analyzed particles in (hk0) reflections[79–81].

**Whole rock analysis**. Whole-rock geochemical analyses of major elements were carried out on 18 biofilms and 23 host sediments at Service d'Analyse des Roches et Minéraux (SARM) of the Centre de Recherches Pétrographiques et Géochimiques (CRPG), Nancy, France. Each sample was powdered in agate mortar and approximately 1 g was fused with lithium metaborate (LiBO₂) and dissolved in nitric acid. Major element concentrations were obtained by inductively-coupled plasma atomic emission spectrometry (ICP-AES).

**Statistics**. For descriptive treatment (boxplot) and statistical analyses, we used the R version 3.5.1[82]. The values for each group do not fulfill the assumption of normality and the Bartlett test of homogeneity of variances was computed prior statistical analyses. Our data returned a significant result (Bartlett test, $\chi^2 = 26.92$, df = 3, p-value < 10⁻⁶), indicating overall inter-group heteroscedasticity of the dataset. According to these results, a multiple factor non-parametric test (i.e., Kruskal–Wallis test) was used to calculate whether the difference in K₂O/SiO₂ ratios is statistically significant or not among lithologies. Finally, post hoc pairwise multiple comparison tests according to Conover were applied to figure out which pairs of groups (i.e., lithologies) are different.

**Carbon analysis**. MRS and their host sediments were powdered for the measurement of organic matter content. We used a FlashEA 1112 (ThermoFisher Scientific) CHNS analyzer for flash dynamic combustion at 970 °C under a constant helium flow. An Eager 300 software was used for data acquisition. A calibration curve was performed with aspartic acid and nicotinamide before each analysis. Carbon content includes both inorganic and organic carbon, but whole-rock geochemical data show that CaO content is low.

**Synchrotron-based scanning X-ray microscopy**. Polished slab sections of representative samples were chosen for scanning XRM analysis. The XRM measurements were performed on several areas within the studied samples at the Nanoscopium hard X-ray nanoprobe beamline[83] of Synchrotron Soleil (L'Orme des Merisiers Saint-Aubin, France). The monochromatic X-ray beam of 12 keV energy was focused on the sample area by a Kirkpatrick–Baez nano-focusing mirror. In order to obtain micrometer resolution for elemental maps of mm²–sized sample areas, we used the fast continuous scanning (FLYSCAN) technique[84]. Full XRF spectra were collected for each pixel of the scans by two Si-drift detectors (VITUS H50, KETEK GmbH) in order to increase the solid angle of detection. The XRF spectra of the two detectors were added and this sum was used for calculating the distributions of S, K, Ca, Ba, Mn, Fe, Ni, Cu, Zn, Ga, Ge, and As in the measured sample regions. Each elemental map was normalized to 10 ms/pixel exposure time.

## Data availability

The authors declare that the data supporting this study are available within the paper and its Supplementary Information files.

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

## Acknowledgements

The authors are grateful to the Gabonese Government, CENAREST, General Direction of Mines and Geology, and Agence Nationale des Parcs Nationaux of Gabon for logistic support. This work was supported by CNRS, the University of Poitiers, Nouvelle Aquitaine Region, and the French Embassy Libreville, Gabon. The authors would like to thank Prof. P. Mouguiama Daouda for his support. The authors also acknowledge E. Chi Fru, B. Dazas, F. Hubert, K. Medjoubi, and L. Pallas for scientific discussions and C. Fontaine, C. Laforest, S. Pronier, and P. Recourt for laboratory support at the Universities of Poitiers and Lille.

## Author contributions

A.E.A. conceived the project. J.A., A.E.A., A.B., and K.O.K. designed research. J.A., A.E.A., O.M.B., and J.-Y.R. performed field research. J.A. performed sample preparation and XRD analyses. A.S. carried out the synchrotron experiments and processed the data. A.M., A.R., R.M., J.-Y.R., A.B., and K.O.K. provided critical input to the manuscript. J.A., A.E.A., A.B., and K.O.K. wrote the manuscript with input from all co-authors.

## Additional information

**Competing interests:** The authors declare no competing interests.

