## [Peer Review File · Nature Communications]

Reviewers' comments:

Reviewer #1 (Remarks to the Author):

The manuscript provides evidence of microbial illitization within pC sediments. The proposed illitization requires a reverse weathering process that the authors posit is driven by active microbial uptake or passive sorption to microbial surfaces. The study compares smectite/illite abundance and crystallinity/crystal size between fossilized microbial mats (both pyritized and unpyritized) and underlying sediments (sandstone and blackshale). Relict microbial mat textures are associated with a.) large crystalline illite crystals, b.) enrichment in potassium, c.) smectite crystals indicating that potassium supply limited illitization. I find the data support the authors' assertions that illite is preferentially associated with microbial mat textures. The mat textures are very convincing and I believe them to be biogenic due to laminations and convincing "escape" structures (e.g. Figure 2). It is not surprising that these mats contain larger illite crystals due to control of diffusive processes within the microbial micro-environment. However, the fact that smectite remains supports the microbial aspect of K-supply proposed by the authors. If biomass is sequestering the K from solution, then incomplete illitization would be expected in the absence of Kspar (which is in fact absent). Overall, this is a really interesting case study into the possibility of microbially-facilitated reverse weathering. I do, however, think that there are a few aspects of the manuscript that could be strengthened.

The weakest piece of data, in my opinion, is the comparison in K enrichment in the pyritized mats. The enrichment is not clear, by any means, especially compared to the non-pyritized samples. The authors suggest that the data demonstrate contamination by the shale. As this is a primary data point it would be appropriate for the authors to pursue this further. An in situ analyses or separation techniques be performed to resolve this issue? Alternatively, is there something about the pyritization process that may interfere with or erase the signature seen in the non-pyritized mat? For example, are we assuming a transition from DIR to DSR (e.g. redox stratification in the sediment column) and if so, would a change in pH (more alkaline to more acidic) impact K sorption? A clearer comparison between the pyritized and non-pyritized might be clear up some of this for readers.

The authors rely on microbially-driven illitization via Fe-reduction from smectite. This makes perfect sense but, as written, this microbial process is detached from the K-supply hypothesis. The manuscript would be strengthened if the authors could weave these processes together into stages of illitization with explicit geochemical reactions and give some context of where or when these reactions occur within the layered microbial community.

Reviewer #2 (Remarks to the Author):

The reviewed manuscript presents a comparison of Paleoproterozoic sediments with microbial biofilms and the corresponding strata sandwiching the biofilms, from a 2.1 Ga old Francevillian section in Gabon. Based on clay mineral and chemical analyses aided by morphological observations, the authors argue that the Paleoproterozoic microbial mats acted as “potassium pump”, sequestering K from sea water and inducing localized illitization during early diagenesis (thus low-temperature). The line of evidence is built on the fact that there is higher K/Si ratio in the biofilms-hosting sediments than in the neighbouring layers. Also, a degree of illitization in the mixed-layered illite-smectite present in the biofilms-associated sediment is suggested to be higher than in other strata of the section. Reaching far with speculative interpretations of bacteria’ role in early Earth’s potassium cycling, the authors, however, did not analyse their data carefully and missed the most obvious and trivial interpretation: there is no potassium enrichment in the biofilms. It is simply a dilution of detrital clay components by quartz, thus lower clay/quartz ratio in the biofilm sediment. Below given is the evidence.

1. Bulk rock chemical composition

K, Al, Mg are common compounds of clay minerals: illite and smectite. Mg and Al are present also in chlorite, which was found in the studied samples. Feldspars (K,Al-; Na-Al-, Ca-Al-bearing) are another compound that often form a detrital assemblage with clays. Albite (Na-feldspar) was identified in the studied rocks. If coming from one source, those minerals can form an assemblage that has constant mineral and chemical ratios, given that Mg is not contributed by carbonate minerals. Such an assemblage can be diluted by detrital quartz from another source, carbonates, organic matter. If K was to be specifically enriched in the biofilms, all other clay-related elements should not be enriched. This is not the case: all clay- and feldspar-related elements are enriched in the non-pyritized mats in respect to silica (Fig. R1). This simply implies that there is nothing specific about K, and all clay+feldspar -related elements are most diluted by quartz in the sandstone (quite obvious) and less diluted in shales or pyritized mats. Accurate mineral XRD quantification performed on correctly prepared bulk samples (see Srodon et al., 2001; Omotoso et al, McCarty et al., and all other Reynolds Cup results) would surely confirm this interpretation.

Figure R1. Element oxides ratios. A dot represents median values, bars – 1 sigma variability.

K and Al form a perfect linear correlation in all four types of studied sediments. The same, with lower r_{sq} , occurs for Mg and Al, implying that Mg comes from clays and not from carbonates (Fig. R2a,b). Only in 2-3 samples from the non-pyritized mats, K and Mg is depleted (!) in respect to Al, whereas Na is enriched (Fig. R2c). This must be the result of increased content of albite. As above, accurate mineral quantification would show this relationship. The linear relationships prove that the clays (and possibly some feldspar) form a common detrital component that is diluted by silica (Fig. R2d).

When compared between different studied sediments, the clay- (and feldspar-) related elements show no statistically significant variability (Fig. R3). As shown already in Fig. R2, they most likely come from the same source, having the same provenance or they underwent hydrodynamical mixing and homogenization.

The reasons why there is less quartz in the non-pyritized mats in respect to other sediments are numerous and most likely trivial. The most obvious is that lower contribution of coarse (quartz-rich) detrital material from the land, thus lower energy deposition allowed development of microbial mats. Alternatively, quartz was initially silica – authigenous in the basin. Regardless the interpretation, it is obvious that detrital clay minerals were associated with biofilms and they were transformed during diagenesis without an additional source of K

Figure R2. Red circles show outliers of the common elemental trend.

Figure R3.

A strong (and real) enrichment in TiO_2 in the non-pyritized mats occurs in respect to both quartz and clays (Fig. R4). Sub-micron size anatase is commonly associated with finest clays due to hydrodynamic sorting. If the concept of the development of a certain type of microbial mats during lowest-energy stages and lowest rate of sedimentation is correct, Ti enrichment is expected as well as lower crystal size of detrital clays.

Figure R4.

2. Clay mineralogy

The authors pursue the idea of illitization during early diagenesis with no proof at all. There is no reference to the overall degree of diagenesis, maximum burial, or maximum paleotemperatures for the section, which would be the first and obvious line of investigation. K-Ar or (encapsulated) Ar-Ar dating, unfortunately, may not help, because the time span between detrital origin, sedimentation and diagenesis can be lower than the analytical error.

The interpretation of clay minerals analysis is far incomplete and unreliable. First, why was the $< 2 \mu\text{m}$ fraction used? Such a fraction represents the major composition of clay minerals in a rock, but not the diagenetic, authigenous portion. With rare exceptions of samples from anchimetamorphism region, such a coarse fraction is dominated by detrital minerals and contains a lot of non-clays – which is clearly the case of the studied samples

(quartz, thick crystallites of 1.0nm-mica, etc.). The detrital components overlap with diagenetic components, making the interpretation unclear. Why was not the finer (< 0.5 , < 0.2 , or even < 0.1 μm) fraction used? Such a fraction is usually quantitatively dominated by diagenetic phases. This has been known for decades; applied and proven in dozens (hundreds?) of papers.

Why no chemical pretreatment was used for clay minerals separation (Jackson, 1956, 1965, and many other papers) to remove at least a part of non-clay minerals observed in the fraction (like gypsum)? There neither was any cation exchange performed, to keep stable interlayer properties for structural determination and comparison among lithological types.

The XRD patterns interpretation does not match standards in clay mineralogy. There is R (order) determined but no % of smectitic layers (%S) in the MLM phase is given. There is no data to compare these values (R and %S) statistically among lithological types (like it was given for bulk chemical composition). There is no clay XRD pattern simulation that would allow quantifying different phases and populations and determining their structural features (%S, R). See tons of papers by Reynolds, Drits, McCarty, Srodon, Lanson, Plancon, Ferrage, Szczerba, Sakharov, etc.

All the above issues are the principles of clay minerals processing and analysis and have been broadly known for decades. It is highly disappointing that the authors did not apply them.

Even if the authors are right about the statistically valid difference in MLM structure between the studied types of sediment, there are numbers of reasons for that, and not related to direct bacterial activity. There are numerous examples of different R and %S in neighboring layers of different lithology; and the reasons vary.

In the paper, the authors present far-reaching conclusions about the role of microbial mats in potassium cycling in the Precambrian. However, extraordinary claims require extraordinary evidence, which was definitely not provided.

There are several minor issues in the submitted manuscript, but discussing them makes no sense in the light of major flaws.

Sincerely,

Arkadiusz (Arek) Derkowski

Reviewer #3 (Remarks to the Author):

Summary: The authors investigated mat-related structures from the Francevillian Formation in southeastern Gabon. They used imaging techniques (TEM, SEM-EDS), bulk XRD, and whole-rock geochemical analyses to identify compositional and textural differences between the mat-related structures (both pyritized and non-pyritized) and background sediments (black shales and sandstones). Large, crystalline illite minerals were found in association with the mat-related structures, but were not commonly found in the background sediment. Higher K₂O/SiO₂ ratios were also found in the mat-related structures relative to the background sediments. Based on these results, the authors conclude that microbes (likely cyanobacteria) played a role in the illitization process by sequestering K⁺, either through uptake into living microbial cells or chelation by EPS.

The results of this study are both interesting and novel, and will interest the broad readership of Nature Communications. To my knowledge, previous works have not investigated K⁺ sequestration by microbial mats. Thus, if published, this study would enhance our knowledge of microbe-mineral interactions, and clay formation.

Major comments:

1. Overall, you need to expand your introduction and discussion sections. Separating your manuscript into separate Introduction/Results/Discussion could help with this. Below are listed a few points/questions that should be addressed in your introduction/discussion:
 - What are other/previous hypotheses for smectite illitization than the one presented in the current study?
 - Smectite illitization mechanisms should be discussed in greater detail. What about the solid-state transformation mechanism (first described by Hower et al. 1976), or the dissolution-recrystallization mechanism?
 - You state the important role of microbial Fe(III) reduction in illitization processes. Can you discuss this further? Why is this an important process in smectite illitization other than “releasing Fe(II) into the sediment pore water”?
2. You conclude that bacteria promote illitization processes. However, smectite illitization has been linked with the presence of organic matter (e.g., Cai et al., 2018*). Thus, are bacteria needed for this process to occur? You allude to the role of bacterial metabolic processes, but have not ruled out alternative scenarios.
3. You should discuss differences between the pyritized and non-pyritized mat patterns, as well as differences between the two background sediments.
4. You should discuss the possible effects of diagenesis. The basin is “unmetamorphosed,” but can you discuss if the illite that is observed in the microbial patterns and background sediment is the original content? Or could it have been subjected to later diagenetic effects? Is the pyrite preserved as original pyrite or an oxidative product, such as iron oxides?

*Cai, J., Du, J., Chen, Z., Lei, T., Zhu, X., 2018. Hydrothermal experiments reveal the influence of organic matter on smectite illitization. *Clays Clay Miner.* 66, 28–42.
doi:10.1346/CCMN.2017.064084

Line-by-line Comments:

Lines 22-41: Typically, citations are not added to manuscript abstracts.

Line 29: “However” should start a new sentence.

Lines 30-34: It is unclear from the way this sentence is worded (and cited) if the discovery of “the high illite content relative to host sediments” has been previously reported, or if this is novel to the current study. I would outline here where the current study differs from the Aubineau et al. 2018 Geobiology paper, e.g., the use of high resolution imaging in the current study.

Line 34-35a: This sentence is confusing; do you mean there were no illite crystals in the background sediment, but these illite crystals were present in the biofilms? Alternatively, you could be describing the relative crystallinity, sizes etc. of the illite minerals present in both the

background sediment and biofilms. Please re-word for clarity.

Line 34-35b: I question whether it is appropriate to use the word "biofilms" here. Instead, consider referring to "mat related structures," as you do later in the manuscript.

Line 37: If the bacteria are trapping K^+ in their EPS, is this necessarily a metabolic activity?

Line 41: You mention applications to Mars science in your abstract, but do not address this again in your main text until the very last sentence. It would be beneficial to elaborate on this in the discussion or remove from the manuscript.

Lines 43-51: You should define illite and smectite before discussing illitization processes. I.e., what types of clays are each of these? How are they different? Where is the location of the K^+ in the illite structure? Differences in interlayer spacing or expandability?

Line 53: Please define or give examples of "mat-growth" and "mat-protected" features.

Line 54: Please provide examples of or define what you mean by "biological activity."

Lines 57-60: You mention textural and mineralogical differences between the host sediment and mat laminae. Please describe what these differences are here. Your next sentence describes MRS textural and compositional features, but does not describe this in the relation to the background sediments.

Lines 69-70: You interpret increases in K_2O/SiO_2 to be indicative of a release in K^+ , but have you considered how fluctuations in silica values would affect this ratio?

Line 94: Please comment on the level of precision in using the stacking mode.

Line 105: What do you mean by "abundances of illite and smectite"? Does this mean total abundance of illite in the sample analyzed? To my knowledge, the stacking method that you used can tell you about %I in the I-S layering, but not absolute abundances of this clay in your sample. If this is correct, this should be stated directly in the paper.

Line 128: This should be "or", "and/or" instead of "and."

Line 204: Explain what you mean by "all lithologies."

Line 205: Were non-pyritized and pyritized samples defined as such before analyses or did you categorize these samples based on XRD results from the current study?

Line 212: Were the $<2 \mu m$ samples powdered before XRD analysis?

Lines 238-239: The SEAD patterns are included in Fig. 4b and Supplementary Fig. 6, but these results are not discussed in the paper. Can you match these SAED results with clay standards?

Figure Comments:

Fig. 3: Please make clear in your figure caption that the background sediment associated with the pyritized mats is the black shales and the background sediment associated with the non-pyritized mats is the sandstones (if I am understanding this figure correctly). In the legend, you show that purple denotes background sediment, but for the sandstones you cannot see that this is purple, making it difficult for the reader to interpret this graph. Furthermore, this graph shows a large discrepancy between K_2O/SiO_2 ratios in pyritized mats relative to non-pyritized mats and black shales relative to sandstones. You should address this in your main text.

Fig. 4b and Supplementary Fig 6: You include the SAED patterns in these images, but need to label these with appropriate index diffraction patterns and Miller indices (hkl).

Reviewer: 1

The manuscript provides evidence of microbial illitization within pC sediments. The proposed illitization requires a reverse weathering process that the authors posit is driven by active microbial uptake or passive sorption to microbial surfaces. The study compares smectite/illite abundance and crystallinity/crystal size between fossilized microbial mats (both pyritized and unpyritized) and underlying sediments (sandstone and black shale). Relict microbial mat textures are associated with a.) large crystalline illite crystals, b.) enrichment in potassium, c.) smectite crystals indicating that potassium supply limited illitization. I find the data support the authors' assertions that illite is preferentially associated with microbial mat textures. The mat textures are very convincing and I believe them to be biogenic due to laminations and convincing "escape" structures (e.g. Figure 2). It is not surprising that these mats contain larger illite crystals due to control of diffusive processes within the microbial micro-environment. However, the fact that smectite remains supports the microbial aspect of K-supply proposed by the authors. If biomass is sequestering the K from solution, then incomplete illitization would be expected in the absence of Kspar (which is in fact absent). Overall, this is a really interesting case study into the possibility of microbially-facilitated reverse weathering. I do, however, think that there are a few aspects of the manuscript that could be strengthened.

The weakest piece of data, in my opinion, is the comparison in K enrichment in the pyritized mats. The enrichment is not clear, by any means, especially compared to the non-pyritized samples. The authors suggest that the data demonstrate contamination by the shale. As this is a primary data point it would be appropriate for the authors to pursue this further. An *in situ* analyses or separation techniques be performed to resolve this issue?

Response: The reviewer is absolutely correct that this is a key piece of missing data. Accordingly, we have now included *in situ* synchrotron analyses that emphasize the local chemical difference in K between the MRS and their hosting sediments (see lines 148-159; Fig. 4; Supplementary Fig. 5). We have now also made statistical analysis on bulk-rock data that highlights difference between pyritized MRS and black shales (see lines 127-130).

Alternatively, is there something about the pyritization process that may interfere with or erase the signature seen in the non-pyritized mat? For example, are we assuming a transition from DIR to DSR (e.g. redox stratification in the sediment column) and if so, would a change in pH (more alkaline to more acidic) impact K sorption?

Response: We can presume that the initial sediments hosted some ferric oxyhydroxides that upon burial would have been reduced via DIR. There was certainly DSR as evident from the pyritized MRS. Both heterotrophic processes lead to alkalinity generation (see below), so increased pH is unlikely to have played a role.

In a related manner, a possible difference between the non-pyritized MRS and the pyritized MRS is that there may have been a shortage of ferric hydroxides in the sediments that became non-pyritized (*i.e.*, pyrite is limited by ferric iron supply). There was clearly organic carbon and sulphate present, and interestingly, the non-pyritized MRS contains more C_{org} (Supplementary Table 1), suggesting that DIR may not have contributed the oxidation of buried biomass. It is

unclear, however, why sulphate reduction would not have consumed all of the organic carbon, unless there was rapid burial out of the sulphate reduction zone. Where the reviewer might be correct is that in the absence of ferric iron to react with dissolved sulphide, the H₂S may have diffused upwards into the overlying oxic sediments where its subsequent oxidation would have generated acidity.

Alternatively, pyritized MRS have been deposited slowly in a low-energy environment, where seawater sulphate would have slowly diffused into sediments, and where DSR would have resulted in sulphidic pore-water conditions close to the sediment-water interface. Thus, reactive Fe would have been reduced via DIR and consumed to form sulphides before smectite-to-illite transformation started. Rapid burial of non-pyritized MRS, which are associated with sandstones and heavy minerals, would not have allowed sulphidic pore-waters to develop. Therefore, reactive iron would have been reduced via DIR and consumed via smectite-to-illite transformation deeper below the sediment-water interface with respect to pyritized MRS.

Given that we really do not know whether the above interpretations are true or not, we would prefer not too much speculate in the text (see lines 117-122).

A clearer comparison between the pyritized and non-pyritized might be clear up some of this for readers.

Response: We have provided additional information on the textural and mineralogical composition that characterized the pyritized and non-pyritized MRS (see lines 101-112).

The authors rely on microbially-driven illitization via Fe-reduction from smectite. This makes perfect sense but, as written, this microbial process is detached from the K-supply hypothesis. The manuscript would be strengthened if the authors could weave these processes together into stages of illitization with explicit geochemical reactions and give some context of where or when these reactions occur within the layered microbial community.

Response: This is a good point. It has been shown that microorganisms can promote the illite-smectite transformation via smectite dissolution by reduction of structural Fe(III) in the smectite structure (e.g., Kim et al., 2004). Considering the decrease of the positive charge in the octahedral sheet as a result of reduced structural Fe(III), an uptake of K into the interlayers may arise to balance the structural charge and produce a neoformed illite. In this study, we argue that the K supply is biologically-derived. We have significantly revised the introduction and discussion to make this clearer (see lines 59-67 and 234-237).

Reviewer: 2

The reviewed manuscript presents a comparison of Paleoproterozoic sediments with microbial biofilms and the corresponding strata sandwiching the biofilms, from a 2.1 Ga old Francevillian section in Gabon. Based on clay mineral and chemical analyses aided by morphological observations, the authors argue that the Paleoproterozoic microbial mats acted as “potassium pump”, sequestering K from sea water and inducing localized illitization during early diagenesis (thus low-temperature). The line of evidence is built on the fact that there is higher K/Si ration in the biofilms-hosting sediments than in the neighbouring layers. Also, a degree of illitization in the mixed-layered illite-smectite present in the biofilms associated sediment is suggested to be higher than in other strata of the section. Reaching far with speculative interpretations of bacteria’ role in early Earth’s potassium cycling, the authors, however, did not analyse their data carefully and missed the most obvious and trivial interpretation: there is no potassium enrichment in the biofilms. It is simply a dilution of detrital clay components by quartz, thus lower clay/quartz ratio in the biofilm sediment. Below given is the evidence.

Response: We understand that the reviewer is sceptical about our interpretations, we certainly should have presented our data more convincingly. In this regard we thank the reviewer for suggesting some excellent tests for our interpretations.

One of the reviewer’s major concerns pertains to quartz dilution. We hope with our new data that he will agree that quartz dilution is not a viable the cause for the K_2O/SiO_2 variations because it cannot explain the observed differences between pyritized and non-pyritized MRS. The pyritized MRS, which have the lowest SiO_2 content (Supplementary Fig. 4a), exhibit lower K_2O/SiO_2 ratio than non-pyritized MRS, inconsistent with the expected quartz dilution effect. Moreover, potassium enrichment in the MRS is consistent with the XRD mineralogical data (Fig. 5a, d; Supplementary Fig. 7), TEM petrographic observations (Figs. 5b, c, e, f; Supplementary Fig. 11), bulk-rock (Fig. 3) and synchrotron geochemical data (Fig. 4; Supplementary Fig. 5). The newly included synchrotron data of the pyritized and non-pyritized MRS, and black shales clearly show K enrichments in the MRS (see lines 148-159).

1. Bulk rock chemical composition

K, Al, Mg are common compounds of clay minerals: illite and smectite. Mg and Al are present also in chlorite, which was found in the studied samples.

Response: The reviewer is correct but the cation content largely determines the specific mineral name of common compounds of clay minerals. For example, chlorites with high Al and Mg contents are called sudoite, and this di-trioctahedral chlorite has been identified in the non-pyritized MRS and sandstones (Supplementary Fig. 6; Billault et al., 2002).

Feldspars (K,Al-, Na-Al-, Ca-Al-bearing) are another compound that often form a detrital assemblage with clays. Albite (Na-feldspar) was identified in the studied rocks.

Response: This might be true elsewhere but the MRS and hosting sediments do not contain any feldspar minerals (Supplementary Fig. 6). Although our previous data suggested the presence of albite in the non-pyritized MRS based on presence of a weak peak ($d= 3.2 \text{ \AA}$), our revised interpretation suggests otherwise (Supplementary Fig. 6). This is because this weak peak at the diagnostic peak position (where it is most intense) is not sufficiently indicative (Brown, 1980),

especially when illite/mica reflections are present (Bailey, 1980). Therefore, we assumed that the weak 3.2 Å peak is illite/mica and not albite in the non-pyritized MRS. Indeed, this is in agreement with the optical and SEM petrographic data, where feldspar was never observed. In addition, this interpretation is supported by the included illite polytypes data of the <2 μm fractions of both MRS and host sediments (see lines 168, 170, and 324-328; Supplementary Fig. 8) that show that the 3.2 Å peak corresponds to the (114) reflection of 2M₁ polytype.

If coming from one source, those minerals can form an assemblage that has constant mineral and chemical ratios, given that Mg is not contributed by carbonate minerals. Such an assemblage can be diluted by detrital quartz from another source, carbonates, organic matter.

Response: We respectfully disagree with the reviewer. We presume he is thinking of methods that are used to characterize the sedimentary provenance with REE and HFSE, which are immobile and do not fractionate during the transport. This approach does not work for major elements where their concentrations in sediments are determined by minerals that host these elements in the provenance, their solubility and resistant to physical weathering. It is well known that sandstones and shales derived from the same provenance would have different chemical composition. K, Na, Mg, and Ca highly soluble in near-surface, low-temperature environments; they are also soluble in low-temperature hydrothermal fluids, resulting in K-metasomatism, for example. Another point is that unless quartz is derived from massive quartz veins, there is no reservoir that can provide only quartz for dilution. We really are unclear here about what the reviewer is suggesting, but there are multiple publications showing that mineral and chemical composition of sediments changes during transport in modern rivers. This effect was even demonstrated for some isotopic ratios (Garcon et al., 2017 for Lu/Hf).

If K was to be specifically enriched in the biofilms, all other clay-related elements should not be enriched. This is not the case: all clay- and feldspar-related elements are enriched in the non-pyritized mats in respect to silica (Fig. R1). This simply implies that there is nothing specific about K, and all clay+feldspar - related elements are most diluted by quartz in the sandstone (quite obvious) and less diluted in shales or pyritized mats.

Response: These plotted ratios argue for quartz dilution (Fig. R1), but we discussed, as shown above, that our results show the contrary. The lack of feldspar suggests that the Al content is only hosted in the phyllosilicates. Because of silica-bearing phyllosilicates we cannot normalized our data with SiO₂ to avoid quartz dilution effect, therefore, K₂O, MgO and Na₂O/Al₂O₃ ratios should be used.

Figure R1. Element oxides ratios. A dot represents median values, bars – 1 sigma variability.

Because feldspars are absent in these sediments, aluminium, magnesium, sodium and potassium are strictly related to phyllosilicates.

Response: Some non-pyritized MRS and sandstones host Al- and Mg- rich chlorite (sudoite), which is totally absent in pyritized MRS and black shales (Supplementary Fig. 6). Explanations are given below for Na enrichment. We agree that clay-related elements are likely diluted by quartz in sandstones. However, silica content does not reflect only quartz, but all silicates, including phyllosilicates.

Accurate mineral XRD quantification performed on correctly prepared bulk samples (see Srodon et al., 2001; Omotoso et al, McCarty et al., and all other Reynolds Cup results) would surely confirm this interpretation.

Response: It is not our goal here to quantify the full clay mineral assemblages. The clay assemblage in non-pyritized MRS is clearly different from those in other lithologies as indicated by SEM and XRD (Figs. 2, 5; Supplementary Figs. 2, 3, 6, 7). The bulk geochemistry data also reflect this difference. We performed XRD quantification using NEWMOD (Reynolds Jr and Reynolds III, 1996), which has widely been used to provide the stacking mode (Reichweite ordering parameter) of I-S MLMs as well as the proportions of smectite and illite layers (*e.g.*, Moore and Reynolds Jr, 1997; Środoń et al., 2009; Gong et al., 2018). Our modelled data confirm the absence of R0 I-S MLMs with an increase in the relative abundance of R3 I-S MLMs in the MRS (see lines 183-189, 315-318, 322, and 333-343; Supplementary Figs. 9, 10; Supplementary Table 3).

K and Al form a perfect linear correlation in all four types of studied sediments.

Response: We provided binary plots of selected major elements (Supplementary Fig. 4) to show their variations within the studied sediments (see lines 131-133). Linear correlations between K and Al are observed in 3 types of the sediments (Supplementary Fig. 4b) but not for the non-pyritized MRS. This is consistent with the presence of K-depleted clay minerals, such as chlorite and sudoite, in the non-pyritized MRS, previously inferred from the XRD results (see lines 139-141). Furthermore, when the data are plotted without normalization to Al_2O_3 (Supplementary Fig. 4), the cross plots show the effect of Si content, but do not account for enrichment or depletion of K. Lastly, the Al_2O_3 -normalized K_2O , MgO and Na_2O plots (Fig. A1 – see below) do not show any variation between the lithologies, indicating major control by the K-, Mg-, and Na-bearing phyllosilicates.

Figure A1. K₂O, MgO and Na₂O/Al₂O₃ ratios.

The same, with lower r_{sq} , occurs for Mg and Al, implying that Mg comes from clays and not from carbonates (Fig. R2a, b).

Response: This is not consistent with what we observed except for sandstones (Supplementary Fig. 4d). Dolomite has been identified in black shales and to a lesser extent in the pyritized MRS by XRD (Supplementary Fig. 6d). The MRS shows no correlation between Mg and Al (Supplementary Fig. 4d), suggesting the presence of Mg-poor clay minerals, such as illite and K-rich I/S MLMs (see lines 143-145), in agreement with XRD.

Only in 2-3 samples from the non-pyritized mats, K and Mg is depleted (!) in respect to Al, whereas Na is enriched (Fig. R2c). This must be the result of increased content of albite.

Response: As we argue above, albite is completely absent in the studied sediments. The low Na₂O content in our sediments (up to 0.29 wt% in black shales) compared to the average shale (0.59 wt%; Li and Schoonmaker, 2003) is consistent with the absence of plagioclase.

Chemical analyses of illitic materials (< 2 μm fraction) reveal that the higher K₂O content of the particles, the higher the Na₂O content (Środoń and Eberl, 1984). This indicates: (1) a correlation of K₂O and Na₂O contents may exist in illite-bearing sediments and (2) illite contain small amounts of Na (see Fig. A2 – below). Therefore, it is not surprising that the MRS exhibit higher K₂O and Na₂O contents (see lines 141-143).

Figure A2: K₂O and Na₂O contents from illitic materials (<2μm fraction) from Środoń and Eberl (1984).

As above, accurate mineral quantification would show this relationship. The linear relationships prove that the clays (and possibly some feldspar) form a common detrital component that is diluted by silica (Fig. R2d). When compared between different studied sediments, the clay- (and feldspar-) related elements show no statistically significant variability (Fig. R3).

Response: We agree, there is no variability between the different studied sediments, and this rather suggests that the selected major elements are dominantly controlled by the same clay species. We note that the Fig. 2d shows very different ratios for pyritized and non-pyritized MRS.

As shown already in Fig. R2, they most likely come from the same source, having the same provenance or they underwent hydrodynamical mixing and homogenization. The reasons why there is less quartz in the non-pyritized mats in respect to other sediments are numerous and most likely trivial. The most obvious is that lower contribution of coarse (quartz-rich) detrital material from the land, thus lower energy deposition allowed development of microbial mats.

Response: This is not consistent with the fact that MRS are on top of several sandstones beds. Indeed, there is a vast amount of literature pertaining to the formation of microbially induced sedimentary structures (MISS), features attributed to microbial mats of filamentous bacteria, such as cyanobacteria, trapping siliciclastic sediments in more energetic tidal flat settings. Indeed, the hydraulic pattern must be moderated because waves and currents in such conditions are strong enough to prohibit huge settlement of fine-grained particles. The filamentous bacteria trapped and bound sediments with their exopolysaccharide (EPS) secretions as they grew upward, and this biostabilization method helped them withstand erosion by waves and currents. The distinctive sedimentary characteristics of MISS can also be found in mats forming in modern tidal flats (e.g., wrinkle structures, roll-up structures, laminated textures) where fossilized equivalents are made up of carbon and iron oxide minerals such as hematite and siderite. Further, cyanobacteria are able to move through sand upward, but they cannot go through mud (Noffke, 2009, 2010). In short, the presence of MISS contradicts the reviewer's statement.

We refer the reviewer to a recent paper of ours in Geobiology (Aubineau et al., 2018) that describes MISS in the same rocks described in this study.

Alternatively, quartz was initially silica – authigenous in the basin. Regardless the interpretation, it is obvious that detrital clay minerals were associated with biofilms and they were transformed during diagenesis without an additional source of K

Reponses: With regards to the suggestion that the silica was authigenic, we are not sure what the reviewer means. There were no silica-secreting organisms at the time of when the Francevillian Group was deposited, so by authigenic we presume he means as a silica cement during early diagenesis. However, our shales do not contain excess of silica with respect to the average shale (Supplementary Table 1; Taylor and McLennan, 2001) (see lines 137-139). Silicification in shales is not common and easily recognized based on physical properties and chemical analyses.

Figure R2. Red circles show outliers of the common elemental trend.

Figure R3.

A strong (and real) enrichment in TiO₂ in the non-pyritized mats occurs in respect to both quartz and clays (Fig. R4). Sub-micron size anatase is commonly associated with finest clays due to hydrodynamic sorting. If the concept of the development of a certain type of microbial mats during lowest-energy stages and lowest rate of sedimentation is correct, Ti enrichment is expected as well as lower crystal size of detrital clays.

Response: The finest clay-size particles were formed during diagenesis, and thus, it is totally inconsistent with hydrodynamic sorting. Non-pyritized MRS are associated with sandstones and have high content of heavy minerals, inconsistent with the environment that the reviewer infers.

Figure R4.

2. Clay mineralogy

The authors pursue the idea of illitization during early diagenesis with no proof at all. There is no reference to the overall degree of diagenesis, maximum burial, or maximum paleotemperatures for the section, which would be the first and obvious line of investigation.

K-Ar or (encapsulated) Ar-Ar dating, unfortunately, may not help, because the time span between detrital origin, sedimentation and diagenesis can be lower than the analytical error.

Response: On lines 247-249, we explain that “microbial respiration of EPS during early diagenesis could liberate adsorbed cations into the pore waters, thus inducing K^+ supersaturation that can additionally promote mineral authigenesis (Dupraz et al., 2009)”. The conversion of smectite to illite-smectite MLMs and then to illite is controlled by different parameters during sediments burial (e.g., Hower et al., 1976; Velde et al., 1986). In addition, we supplied additional information on the diagenetic history of the Francevillian FB Formation (see lines 70-73).

The interpretation of clay minerals analysis is far incomplete and unreliable. First, why was the $< 2 \mu\text{m}$ fraction used? Such a fraction represents the major composition of clay minerals in a rock, but not the diagenetic, authigenous portion.

Response: The reviewer’s statement is simply not true. In our study, the $<2 \mu\text{m}$ fraction revealed illite-smectite MLMs, which formed during diagenesis (Środoń, 1979, 1984; Velde et al., 1986; Lanson et al., 2009, among others). The distribution and evolution of the illite-smectite MLMs minerals in the FB Formation were previously discussed in Ossa Ossa et al. (2013) and Ngombi-Pemba et al. (2014), and these authors clearly modelled the diagenetic nature of different size fractions. Moreover, the $<2 \mu\text{m}$ fraction have been used by several authors to characterize various diagenetic clay minerals (e.g., Środoń and Eberl, 1984; Drits, 1993; Inoue and Kitagawa, 1994; Billault et al., 2002; Lanson et al., 2002; Haines and van der Pluijm, 2008; Bankole et al., 2015, 2016). The illite polytypes of the $<2 \mu\text{m}$ fraction of randomly oriented powders (Supplementary Figure 8) show that our samples are dominantly composed of mixtures of $1M_t$ and $2M_1$ polytypes without any trace of $1M_c$ polytype (Bailey, 1980; Drits, 1993; Haines and van der Pluijm, 2008). $1M_t$ are neoformed illites while $2M_1$ could be either neoformed or inherited (Grathoff et al., 2001).

With rare exceptions of samples from anchimetamorphism region, such a coarse fraction is dominated by detrital minerals and contains a lot of non-clays – which is clearly the case of the studied samples (quartz, thick crystallites of 1.0nm-mica, etc.).

Response: Our observations revealed the presence of lath-shaped and hexagonal clay particles that are typically formed during burial diagenesis (see lines 194-197; Figs. 5b, c, e, f; Supplementary Fig. 11) (Nadeau et al., 1985; Eberl and Środoń, 1988; Inoue et al., 1988; Whitney and Velde, 1993; Inoue and Kitagawa, 1994).

The detrital components overlap with diagenetic components, making the interpretation unclear. Why was not the finer (< 0.5 , < 0.2 , or even $< 0.1 \mu\text{m}$) fraction used? Such a fraction is usually quantitatively dominated by diagenetic phases. This has been known for decades; applied and proven in dozens (hundreds?) of papers.

Response: We agree that finer fractions are dominated by diagenetic phases, but as discussed above diagenetic phases have also been recognized in the $<2 \mu\text{m}$ clay fraction.

Why no chemical pretreatment was used for clay minerals separation (Jackson, 1956, 1965, and many other papers) to remove at least a part of non-clay minerals observed in the fraction (like gypsum)?

Response: Gypsum does not affect the clay mineral separation nor interfere with its identification on XRD profiles.

There neither was any cation exchange performed, to keep stable interlayer properties for structural determination and comparison among lithological types.

Response: The samples are Ca-saturated and analysed in AD and EG states for profile fitting process (see lines 315-318 and 322). Layer charge determination is of no importance in this study.

The XRD patterns interpretation does not match standards in clay mineralogy. There is R (order) determined but no % of smectitic layers (%S) in the MLM phase is given. There is no data to compare these values (R and %S) statistically among lithological types (like it was given for bulk chemical composition). There is no clay XRD pattern simulation that would allow quantifying different phases and populations and determining their structural features (%S, R). See tons of papers by Reynolds, Drits, McCarty, Srodon, Lanson, Plancon, Ferrage, Szczerba, Sakharov, etc. All the above issues are the principles of clay minerals processing and analysis and have been broadly known for decades. It is highly disappointing that the authors did not apply them.

Response: The reviewer is correct. As a consequence, we have revised the interpretation of the XRD patterns (see lines 167-171 and 175-182; Figs 5a, d; Supplementary Fig. 7). We also carried out XRD pattern simulations (Supplementary Figs 9-10; Supplementary Table 3) and the simulation results allowed us to identify R3 I-S MLMs with more than 88% of illite in both MRS and sediments. The relative abundance of such MLMs increases within the MRS and no corresponding increase in R0 I-S MLMs was observed (see lines 183-189). The latter are only found in the hosting sediment with 65% of smectite.

Even if the authors are right about the statistically valid difference in MLM structure between the studied types of sediment, there are numbers of reasons for that, and not related to direct bacterial activity. There are numerous examples of different R and %S in neighboring layers of different lithology; and the reasons vary.

Response: To be honest, this comment is not helpful unless those examples are provided. We discussed the sources of difference and feel that our interpretation corresponds best to the data we have. If the reviewer has an alternative interpretation, we would like to hear it.

In the paper, the authors present far-reaching conclusions about the role of microbial mats in potassium cycling in the Precambrian. However, extraordinary claims require extraordinary evidence, which was definitely not provided. There are several minor issues in the submitted manuscript, but discussing them makes no sense in the light of major flaws.

Sincerely,
Arkadiusz (Arek) Derkowski

Reviewer: 3

Summary: The authors investigated mat-related structures from the Francevillian Formation in southeastern Gabon. They used imaging techniques (TEM, SEM-EDS), bulk XRD, and whole-rock geochemical analyses to identify compositional and textural differences between the mat-related structures (both pyritized and non-pyritized) and background sediments (black shales and sandstones). Large, crystalline illite minerals were found in association with the mat-related structures, but were not commonly found in the background sediment. Higher K₂O/SiO₂ ratios were also found in the mat-related structures relative to the background sediments. Based on these results, the authors conclude that microbes (likely cyanobacteria) played a role in the illitization process by sequestering K⁺, either through uptake into living microbial cells or chelation by EPS.

The results of this study are both interesting and novel, and will interest the broad readership of Nature Communications. To my knowledge, previous works have not investigated K⁺ sequestration by microbial mats. Thus, if published, this study would enhance our knowledge of microbe-mineral interactions, and clay formation.

Response: We sincerely thank the reviewer for these positive comments.

Major comments:

1. Overall, you need to expand your introduction and discussion sections. Separating your manuscript into separate Introduction/Results/Discussion could help with this. Below are listed a few points/questions that should be addressed in your introduction/discussion:

- What are other/previous hypotheses for smectite illitization than the one presented in the current study?

Response: We have expanded the introduction, and also provided additional information on the hypotheses as suggested (see lines 59-67).

- Smectite illitization mechanisms should be discussed in greater detail. What about the solid-state transformation mechanism (first described by Hower et al. 1976), or the dissolution-recrystallization mechanism?

Response: As suggested, we now better explain the smectite-illitization mechanisms (see lines 49-58).

- You state the important role of microbial Fe(III) reduction in illitization processes. Can you discuss this further? Why is this an important process in smectite illitization other than “releasing Fe(II) into the sediment pore water”?

Response: As also suggested by reviewer #1, we have revised this aspect and added more information in the introduction and discussion (see lines 60-65 and 234-237). Microorganisms can promote the illite-smectite transformation via the smectite dissolution by reduction of structural Fe(III) in the smectite structure (Kim et al., 2004). Considering the decrease of the positive charge in the octahedral sheet due to reduction of structural Fe(III), a K uptake into the interlayers may arise to balance the charge, and thus neoforining illite. In this study, we argue that the K supply is biologically-derived.

2. You conclude that bacteria promote illitization processes. However, smectite illitization has been linked with the presence of organic matter (e.g., Cai et al., 2018*). Thus, are bacteria needed for this process to occur? You allude to the role of bacterial metabolic processes, but have not ruled out alternative scenarios.

Response: Cai et al. (2018) reveal that organic matter influences smectite illitization under hydrothermal experiments. Such conditions are not reported in the Francevillian FB Formation (El Albani et al., 2010, 2014). Organic matter does influence smectite illitization by delaying the exchange of interlayer cations (Li et al., 2016). However, in our study, non-pyritized MRS have both high organic carbon contents (Supplementary Table 1) and an advanced stage of illitisation. Therefore, we have suggested that there is no link between organic matter and smectite illitisation in our samples and linked illitisation to bacterial metabolic processes (see lines 225-228).

3. You should discuss differences between the pyritized and non-pyritized mat patterns, as well as differences between the two background sediments.

Response: Agreed, and additional information has now been provided (see lines 77-79, 101-112, and 117-122).

4. You should discuss the possible effects of diagenesis. The basin is “unmetamorphosed,” but can you discuss if the illite that is observed in the microbial patterns and background sediment is the original content? Or could it have been subjected to later diagenetic effects? Is the pyrite preserved as original pyrite or an oxidative product, such as iron oxides?

Response: The Francevillian sedimentary rocks have undergone a common burial diagenesis (see lines 70-73). The main parameters that control the smectite to illite-smectite reaction are K^+ availability and time (e.g., Velde et al., 1986). The ~ 2.1 billion-years-old underlying FA Formation contains K-Feldspars with the dominance of diagenetic illite polytypes without illite-smectite MLMs, suggesting advanced stage of illitization (Bankole et al., 2015). On the other hand, sediments from the FB Formation lack K-feldspars and the completeness of illitisation is not achieved (Ngombi-Pemba et al., 2014). Locally, an advanced stage of illitisation is observed in K-bentonite (Bankole et al., 2018) and in MRS (this study) in the FB Formation. Therefore, the illite content in these sediments largely depends on the local availability of potassium. At the MRS levels, illite crystal habits revealed by TEM (Figs. 5b, c, e, f; Supplementary Fig. 11) show that these particles are neoformed (lath- or hexagonal-shaped) as usually observed in diagenetic series (Eberl and Środoń, 1988; Inoue et al., 1988; Inoue and Kitagawa, 1994).

In addition, the XRD patterns indicate that iron oxides are completely absent in the studied sediments. Pyrite is the only iron sulphide observed in the FB Formation and mostly occur as isolated concretions as commonly observed in black shales. However, the small crystals, which are disseminated inside MRS do not exhibit any overgrowth (SEM observations) that could be due to late diagenetic processes.

Line-by-line Comments:

Lines 22-41: Typically, citations are not added to manuscript abstracts.

Response: We removed citations (see lines 24-47).

Line 29: “However” should start a new sentence.

Response: We added “however” (see line 31).

Lines 30-34: It is unclear from the way this sentence is worded (and cited) if the discovery of “the high illite content relative to host sediments” has been previously reported, or if this is novel to the current study. I would outline here where the current study differs from the Aubineau et al. 2018 Geobiology paper, e.g., the use of high resolution imaging in the current study.

Response: We removed “the high illite content relative to host sediments” for clarity. This is an original contribution and these observations have not been previously described in Aubineau et al. (2018).

Line 34-35a: This sentence is confusing; do you mean there were no illite crystals in the background sediment, but these illite crystals were present in the biofilms? Alternatively, you could be describing the relative crystallinity, sizes etc. of the illite minerals present in both the background sediment and biofilms. Please re-word for clarity.

Response: We rewrote the sentence (see lines 36-38). Illitic particles are present in the hosting sediments. However, they are more abundant and have larger size in the MRS due to the advanced stage of illitisation reaction. Such textures suggest the effect of the Ostwald ripening-like process as typically described in diagenetic series (Eberl and Środoń, 1988).

Line 34-35b: I question whether it is appropriate to use the word “biofilms” here. Instead, consider referring to “mat related structures,” as you do later in the manuscript.

Response: We used MRS instead of biofilms for consistency (see line 43).

Line 37: If the bacteria are trapping K^+ in their EPS, is this necessarily a metabolic activity?

Response: This is a good question. The reviewer is correct that this is not necessarily metabolic activity, and as such we have reworded the text (see line 42). What we should have stressed is that it is an example of microbial activity because without the microbes in the first place, there would be no EPS to sequester the K^+ .

Line 41: You mention applications to Mars science in your abstract, but do not address this again in your main text until the very last sentence. It would be beneficial to elaborate on this in the discussion or remove from the manuscript.

Response: We removed the Mars aspect as the reviewer is correct, it is not our primary focus (see lines 47 and 288).

Lines 43-51: You should define illite and smectite before discussing illitization processes. I.e., what types of clays are each of these? How are they different? Where is the location of the K^+ in the illite structure? Differences in interlayer spacing or expandability?

Response: Smectite and illite are 2:1 phyllosilicates which differ by their capacity to absorb water or organic molecules such as ethylene glycol in their interlayer regions. Smectite is a swelling mineral and expands in the presence of water or organic molecules, while illite is a non-swelling mineral due to the presence of potassium cations in the interlayer space.

Line 53: Please define or give examples of “mat-growth” and “mat-protected” features.

Response: These terms have previously been described in Aubineau et al. (2018) and follow the classification scheme of Sarkar et al. (2008). However, the reviewer is correct and the reader should not have to refer to another reference to obtain this information. Accordingly, we have now defined those terms (see lines 88-94).

Line 54: Please provide examples of or define what you mean by “biological activity.”

Response: Biological activity, as used in this study, means undirected gliding motility and/or phototactic behaviour (see lines 91-93) (Shepard and Sumner, 2010; Reyes et al., 2013).

Lines 57-60: You mention textural and mineralogical differences between the host sediment and mat laminae. Please describe what these differences are here. Your next sentence describes MRS textural and compositional features, but does not describe this in the relation to the background sediments.

Response: We have expanded description of difference between the MRS and hosting sediments (see lines 101-112).

Lines 69-70: You interpret increases in K_2O/SiO_2 to be indicative of a release in K^+ , but have you considered how fluctuations in silica values would affect this ratio?

Response: This was also the main concern of the reviewer#2. Hence, we decided to incorporate binary plots of selected major elements (see lines 131-133; Supplementary Fig. 4) in an effort to provide data to evaluate elemental variations amongst all rock types. While it is true that some quartz dilution is evident in our sediments – from direct correlation with SiO_2 contents using bulk-rock geochemistry; the difference between pyritized and non-pyritized MRS cannot be explained by this process. Pyritized MRS have the lowest SiO_2 content and lower K_2O/SiO_2 ratio than non-pyritized MRS. Hence, quartz dilution does not explain K enrichment (see lines 133-137). We have also included *in situ* synchrotron analyses on pyritized and non-pyritized MRS, and black shales (Fig. 4; Supplementary Fig. 5) to further prove K enrichment within the MRS (see lines 148-159).

Line 94: Please comment on the level of precision in using the stacking mode.

Response: The stacking mode and the I-S ratio are determined by the position and intensity of the diffraction bands. Using profile calculation (NEWMOD program simulation; Moore and Reynolds Jr., 1997), it is possible to fit the experimental pattern - the better the fit the higher the precision.

Line 105: What do you mean by “abundances of illite and smectite”? Does this mean total abundance of illite in the sample analyzed? To my knowledge, the stacking method that you used can tell you about %I in the I-S layering, but not absolute abundances of this clay in your sample. If this is correct, this should be stated directly in the paper.

Response: Abundance, as used here, refers to the percentage of illite layers in the I-S mixed layering and not the absolute amounts of smectite or illite in the sediment or the MRS. Consequently, we observed that the relative abundance of illite particles is significantly higher in MRS than in their hosting sediments. The simulated results (Supplementary Figs 9-10;

Supplementary Table 3) allowed us to identify R3 I-S MLMs with more than 88% of illite in both MRS and hosting sediments. However, the relative abundance of such MLMs increases within the MRS and, no R0 I-S MLMs has been recognized in this lithology (see lines 183-189).

Line 128: This should be “or”, “and/or” instead of “and.”

Response: We changed “and” by “and/or (see line 242).

Line 204: Explain what you mean by “all lithologies.”

Response: All lithologies refer to the four categories of the studied sediments: pyritized and non-pyritized MRS, black shale, and sandstone (see line 298).

Line 205: Were non-pyritized and pyritized samples defined as such before analyses or did you categorize these samples based on XRD results from the current study?

Response: A combination of results of petrographic (Aubineau et al., 2018) and sulphur contents acquired by whole-rock geochemistry analyses allowed us to distinguish the two MRS types. In this paper, we added specific information on how the MRS distinction was made: sulphur content (see lines 299-300) and XRD analyses. Besides, the pyritized MRS are in the black shales, but are lacking along the bedding planes of sandstones, whereas the non-pyritized MRS are strictly restricted to the sandstone facies.

Line 212: Were the $<2 \mu\text{m}$ samples powdered before XRD analysis?

Response: First, the $<2 \mu\text{m}$ clay size particles were separated by dispersion of bulk sample material in deionized water, allowed to freely settle under gravity at a controlled room temperature of 20°C , and centrifuged to separate $<2 \mu\text{m}$ clay fraction. The separated $<2 \mu\text{m}$ solution was thereafter pipetted on glass slide and dried at room temperature for XRD analysis.

Lines 238-239: The SEAD patterns are included in Fig. 4b and Supplementary Fig. 6, but these results are not discussed in the paper. Can you match these SAED results with clay standards?

Response: The SAED patterns reveal the stacking arrangement of the layers (Veblen et al., 1990; Whitney and Velde, 1993; Kim et al., 2004). A turbostratic ring pattern is typical of turbostratic smectite structure (random rotations between layers), while non-turbostratic (*i.e.*, coherent) stacking pattern is commonly observed in illite crystal structure (hk0 dots) forming a hexagonal pattern (Kumai, 1976). Thus, in this study, we used TEM to observe and identify the crystal habits of the illite particles. More information has been provided (see lines 356-358).

Figure Comments:

Fig. 3: Please make clear in your figure caption that the background sediment associated with the pyritized mats is the black shales and the background sediment associated with the non-pyritized mats is the sandstones (if I am understanding this figure correctly). In the legend, you show that purple denotes background sediment, but for the sandstones you cannot see that this is purple, making it difficult for the reader to interpret this graph. Furthermore, this graph shows a large discrepancy between $\text{K}_2\text{O}/\text{SiO}_2$ ratios in pyritized mats relative to non-pyritized mats and black shales relative to sandstones. You should address this in your main text.

Response: We have now clarified the figure caption (see lines 796-802). We have also changed the colour of the box borders to make the figure clearer. Our main interest is to show the difference in K_2O/SiO_2 in the MRS relative to their hosting sediments, but not to compare all rock types because of possible quartz dilution effect.

Fig. 4b and Supplementary Fig 6: You include the SAED patterns in these images, but need to label these with appropriate index diffraction patterns and Miller indices (hkl).

Response: We labelled the SAED patterns (see lines 813 and 875). They all show $hk0$ patterns. Kumai (1976) provided the indexing of planes having indices ($hk0$) of electron diffraction pattern and this is applicable for all orthohexagonal crystals.

References:

- Aubineau, J. et al., 2018, Unusual microbial mat-related structural diversity 2.1 billion years ago and implications for the Francevillian biota: *Geobiology*, v. 16, p. 476–497, doi:10.1111/gbi.12296.
- Bailey, S.W., 1980, Structure of layer silicates, *in* Brindley, G.W. and Brown, G. eds., *Crystal structure of clay minerals and their X-Ray identification*, London, Mineralogical Society, p. 1–123.
- Bankole, O.M., El Albani, A., Meunier, A., and Gauthier-Lafaye, F., 2015, Textural and paleo-fluid flow control on diagenesis in the Paleoproterozoic Franceville Basin, South Eastern, Gabon: *Precambrian Research*, v. 268, p. 115–134, doi:10.1016/j.precamres.2015.07.008.
- Bankole, O.M., El Albani, A., Meunier, A., Pambo, F., Paquette, J.-L., and Bekker, A., 2018, Earth's oldest preserved K-bentonites in the *ca.* 2.1 Ga Francevillian Basin, Gabon: *American Journal of Science*, v. 318, p. 409–434, doi:10.2475/04.2018.02.
- Bankole, O.M., El Albani, A., Meunier, A., Rouxel, O.J., Gauthier-Lafaye, F., and Bekker, A., 2016, Origin of red beds in the Paleoproterozoic Franceville Basin, Gabon, and implications for sandstone-hosted uranium mineralization: *American Journal of Science*, v. 316, p. 839–872.
- Billault, V., Beaufort, D., Patrier, P., and Petit, S., 2002, Crystal chemistry of Fe-sudoites from uranium deposits in the Athabasca basin (Saskatchewan, Canada): *Clays and Clay Minerals*, v. 50, p. 70–81.
- Brown, G., 1980, Associated minerals, *in* Brindley, G.W. and Brown, G. eds., *Crystal structure of clay minerals and their X-Ray identification*, London, Mineralogical Society, p. 361–410.
- Drits, V.A., 1993, X-Ray Identification of One-Layer Illite Varieties: Application to the Study of Illites around Uranium Deposits of Canada: *Clays and Clay Minerals*, v. 41, p. 389–398, doi:10.1346/CCMN.1993.0410316.
- Dupraz, C., Reid, R.P., Braissant, O., Decho, A.W., Norman, R.S., and Visscher, P.T., 2009, Processes of carbonate precipitation in modern microbial mats: *Earth-Science Reviews*, v. 96, p. 141–162, doi:10.1016/j.earscirev.2008.10.005.
- Eberl, D.D., and Šrodoň, J., 1988, Ostwald ripening and interparticle-diffraction effects for illite crystals: *American Mineralogist*, v. 73, p. 1335–1345.
- El Albani, A.E. et al., 2010, Large colonial organisms with coordinated growth in oxygenated environments 2.1 Gyr ago: *Nature*, v. 466, p. 100–104, doi:10.1038/nature09166.
- El Albani, A. et al., 2014, The 2.1 Ga Old Francevillian Biota: Biogenicity, Taphonomy and Biodiversity (L. Rook, Ed.): *PLoS ONE*, v. 9, p. e99438, doi:10.1371/journal.pone.0099438.
- Garçon, M., Carlson, R.W., Shirey, S.B., Arndt, N.T., Horan, M.F., and Mock, T.D., 2017, Erosion of Archean continents: The Sm-Nd and Lu-Hf isotopic record of Barberton sedimentary rocks: *Geochimica et Cosmochimica Acta*, v. 206, p. 216–235, doi:10.1016/j.gca.2017.03.006.
- Gong, N., Hong, H., Huff, W.D., Fang, Q., Bae, C.J., Wang, C., Yin, K., and Chen, S., 2018, Influences of Sedimentary Environments and Volcanic Sources on Diagenetic Alteration of Volcanic Tuffs in South China: *Scientific Reports*, v. 8, doi:10.1038/s41598-018-26044-w.
- Grathoff, G.H., Moore, D.M., Hay, R.L., and Wemmer, K., 2001, Origin of illite in the lower Paleozoic of the Illinois basin: Evidence for brine migrations: *Geological Society of America Bulletin*, v. 113, p. 1092–1104.
- Haines, S.H., and van der Pluijm, B.A., 2008, Clay quantification and Ar–Ar dating of

synthetic and natural gouge: Application to the Miocene Sierra Mazatán detachment fault, Sonora, Mexico: *Journal of Structural Geology*, v. 30, p. 525–538, doi:10.1016/j.jsg.2007.11.012.

Hower, J., Eslinger, E.V., Hower, M.E., and Perry Jr, E.A., 1976, Mechanism of burial metamorphism of argillaceous sediment: 1. Mineralogical and chemical evidence: *Geological Society of America Bulletin*, v. 87, p. 725, doi:10.1130/0016-7606(1976)87<725:MOBMOA>2.0.CO;2.

Inoue, A., and Kitagawa, R., 1994, Morphological characteristics of illitic clay minerals from a hydrothermal system: *American Mineralogist*, v. 79, p. 700–711.

Inoue, A., Velde, B., Meunier, A., and Touchard, G., 1988, Mechanism of illite formation during smectite-to-illite conversion in a hydrothermal system: *American Mineralogist*, v. 73, p. 1325–1334.

Kim, J., Dong, H., Seabaugh, J., Newell, S.W., and Eberl, D.D., 2004, Role of microbes in the smectite-to-illite reaction: *Science*, v. 303, p. 830–832.

Kumai, M., 1976, Identification of Nuclei and concentrations of chemical species in snow crystals sampled at the South Pole: *Journal of the Atmospheric Sciences*, v. 33, p. 833–841.

Lanson, B., Beaufort, D., Berger, G., Bauer, A., Cassagnabère, A., and Meunier, A., 2002, Authigenic kaolin and illitic minerals during burial diagenesis of sandstones: a review: *Clay Minerals*, v. 37, p. 1–22, doi:10.1180/0009855023710014.

Lanson, B., Sakharov, B.A., Claret, F., and Drits, V.A., 2009, Diagenetic smectite-to-illite transition in clay-rich sediments: a reappraisal of X-Ray diffraction results using the multi-specimen method: *American Journal of Science*, v. 309, p. 476–516.

Li, Y., Cai, J., Song, M., Ji, J., and Bao, Y., 2016, Influence of organic matter on smectite illitization: A comparison between red and dark mudstones from the Dongying Depression, China: *American Mineralogist*, v. 101, p. 134–145, doi:10.2138/am-2016-5263.

Li, Y.H., and Schoonmaker, J., 2003, 7.01 Chemical composition and mineralogy of marine sediments, *in* Mackenzie, F.T. ed., *Treatise on Geochemistry*, Vol. 7: Sediments, Diagenesis, and Sedimentary Rocks, Oxford, Elsevier, p. 1–35.

Moore, D.M., and Reynolds Jr, R.C., 1997, X-ray diffraction and the identification and analysis of clay minerals: New York, Oxford University Press, 378 p.

Nadeau, P.H., Wilson, M.J., McHardy, W.J., and Tait, J.M., 1985, The conversion of smectite to illite during diagenesis: evidence from some illitic clays from bentonites and sandstones: *Mineralogical Magazine*, v. 49, p. 393–400, doi:10.1180/minmag.1985.049.352.10.

Ngombi-Pemba, L., Albani, A.E., Meunier, A., Grauby, O., and Gauthier-Lafaye, F., 2014, From detrital heritage to diagenetic transformations, the message of clay minerals contained within shales of the Palaeoproterozoic Francevillian basin (Gabon): *Precambrian Research*, v. 255, p. 63–76, doi:10.1016/j.precamres.2014.09.016.

Noffke, N., 2010, *Geobiology*: Berlin, Heidelberg, Springer Berlin Heidelberg, doi:10.1007/978-3-642-12772-4.

Noffke, N., 2009, The criteria for the biogenicity of microbially induced sedimentary structures (MISS) in Archean and younger, sandy deposits: *Earth-Science Reviews*, v. 96, p. 173–180, doi:10.1016/j.earscirev.2008.08.002.

Ossa Ossa, F. et al., 2013, Exceptional preservation of expandable clay minerals in the ca. 2.1Ga black shales of the Francevillian basin, Gabon and its implication for atmospheric oxygen accumulation: *Chemical Geology*, v. 362, p. 181–192, doi:10.1016/j.chemgeo.2013.08.011.

Reyes, K., Gonzalez, N.I., Stewart, J., Ospino, F., Nguyen, D., Cho, D.T., Ghahremani, N., Spear, J.R., and Johnson, H.A., 2013, Surface Orientation Affects the Direction of Cone

Growth by *Leptolyngbya* sp. Strain C1, a Likely Architect of Coniform Structures Octopus Spring (Yellowstone National Park): *Applied and Environmental Microbiology*, v. 79, p. 1302–1308, doi:10.1128/AEM.03008-12.

Reynolds Jr, R.C., and Reynolds III, R.C., 1996, NEWMOD for Windows. The calculation of one-dimensional X-ray diffraction patterns of mixed-layered clay minerals: 8 Brook Road, Hanover, New Hampshire.

Sarkar, S., Bose, P., Samanta, P., Sengupta, P., and Eriksson, P., 2008, Microbial mat mediated structures in the Ediacaran Sonia Sandstone, Rajasthan, India, and their implications for Proterozoic sedimentation: *Precambrian Research*, v. 162, p. 248–263, doi:10.1016/j.precamres.2007.07.019.

Shepard, R.N., and Sumner, D.Y., 2010, Undirected motility of filamentous cyanobacteria produces reticulate mats: *Motility produces reticulate mats: Geobiology*, v. 8, p. 179–190, doi:10.1111/j.1472-4669.2010.00235.x.

Środoń, J., 1979, Correlation Between Coal and Clay Diagenesis in the Carboniferous of the Upper Silesian Coal Basin, *in* Mortland, M.M. and Farmer, V.C. eds., *Developments in Sedimentology*, Elsevier, v. 27, p. 251–260, doi:10.1016/S0070-4571(08)70721-0.

Środoń, J., 1984, Mixed-layer illite-smectite in low-temperature diagenesis: data from the Miocene of the Carpathian Foredeep: *Clay Minerals*, v. 19, p. 205–215, doi:10.1180/claymin.1984.019.2.07.

Środoń, J., and Eberl, D.D., 1984, Illite, *in* Bailey, S.W. ed., *Review in Mineralogy* 13, Micas, Washington DC, Mineralogical Society of America, p. 495–544.

Środoń, J., Zeelmaekers, E., and Derkowski, A., 2009, The charge of component layers of illite-smectite in bentonites and the nature of end-member illite: *Clays and Clay Minerals*, v. 57, p. 649–671, doi:10.1346/CCMN.2009.0570511.

Taylor, S.R., and McLennan, S.M., 2001, Chemical Composition and Element Distribution in the Earth's Crust, *in* Roberts, A.M. ed., *Encyclopedia of Physical Science and Technology*, New York, Academic Press, p. 697–719.

Veblen, D.R., Gutherie Jr, G.D., Livi, K.J.T., and Reynolds Jr, R.C., 1990, High-Resolution Transmission Electron Microscopy and Electron Diffraction of Mixed-Layer Illite/Smectite: Experimental Results: *Clays and Clay Minerals*, v. 38, p. 1–13, doi:10.1346/CCMN.1990.0380101.

Velde, B., Suzuki, T., and Nicot, E., 1986, Pressure-temperature-composition of illite/smectite mixed-layer minerals: Niger delta mudstones and other examples: *Clays and Clay Minerals*, v. 34, p. 435–441.

Whitney, G., and Velde, B., 1993, Changes in Particle Morphology During Illitization: An Experimental Study: *Clays and Clay Minerals*, v. 41, p. 209–218, doi:10.1346/CCMN.1993.0410209.

REVIEWERS' COMMENTS:

Reviewer #1 (Remarks to the Author):

The authors provide a compelling example of microbial influence on early diagenesis, particularly related to clay diagenesis. This is a novel contribution that should promote interest and further study. The new data and revised text support a role for microbial contributions to reverse weathering processes.

Importantly, the authors have addressed my concerns regarding K concentrations with the addition of synchrotron data. They did further diligence in addressing reviewer 2's (and reviewer 3) concerns about quartz dilution and I am satisfied with the support for their interpretation. The added information regarding the role of different microbial metabolisms, particularly as they relate to illitization, make these complex processes clear to the reader and more accessible to those scientists who are not as familiar with the particulars of environmental microbiology.

This is a well supported and well written submission that I hope to see published due to its novel insight into microbe: mineral interactions

in addition to its strong approach to interpreting those processes in ancient rock.

Reviewer #3 (Remarks to the Author):

This is a much improved version of the original submission. Major comments* include the following:

- (1) Please pay attention to grammar/clarity issues.
- (2) Please remove all discussion from your results section.
- (3) There are several results that are not addressed in your discussion section, including your TEM data and elemental ratio plots. Please be sure to discuss the implications of all results.

(4) Your evidence for K⁺ uptake by microbial communities is convincing. However, the broader implications may require additional thought and discussion.

(5) Please address issues with XRF figures.

*All general comments are more thoroughly explained/detailed in the line-by-line comments (below).

Line-by-line comments:

Line 1: The current paper does not describe clay formation, but rather the transformation from smectite to illite-rich clay layers. A reference to illitization or partial illitization (versus clay formation) would better describe this research.

Line 24: Please remove the parentheses and incorporate the definition of illitization into the first sentence.

Line 25: Replace “the smectite” with “a smectite.”

Line 26-27a: What do you mean here by “sediment pile”? You have not discussed sediment previously, only clay. Do you mean to say the clay-rich sediment on which the bacteria are living?

Line 26-27b: Also, is there a difference between “uptake” and “supply” to the sediment? If these are different processes, please elaborate. If not, then please remove one of the two.

Line 27c: You say “whether illitization was equally important in the geologic past.” Please clarify this statement. i.e., “equally important” to what?

Line 28: Replace “2.1 billion-years” with “2.1-billion-year-old.”

Line 31: Please elaborate here on what you mean by “in the MRS.” Do you mean physically associated with the filaments?

Line 32: I am unfamiliar with the term “hosting sediments.” Do you mean “host sediments?” If so, please replace here and elsewhere in the manuscript.

Line 33: Replace “released it in the pore-waters” with “released it into the pore waters.”

Lines 34-38: I find this sentence to be a bit confusing. Please re-word for clarity and check for grammar issues.

Line 41: Remove “it” from this sentence.

Line 43-44: This entire sentence needs to be re-worded for clarity. Firstly, you mean to characterize the “illite-rich layers” in this sentence, rather than the “smectite-rich layers.” By simply adding something to the effect of “the latter of which is characterized by...” this would clear up the first point of confusion as to what exactly you are characterizing. Secondly, if I am reading this sentence correctly, you mean to say that the illite-rich layers are composed of “interstratified I-S MLMs and long-range ordered I-S MLMs”, i.e., the illite-rich layers are not comprised of 100% illite, and this only represents a partial illitization process. However, you add that these are actually “smectite-rich layers” as well; please clarify how layers can be both illite- and smectite-rich.

Line 44-46: Add references regarding the “multiple mechanisms.”

Line 46: Why are these proposed illitization mechanisms considered to be controversial? You say so here, but then only two give examples of mechanisms, and do not explain the controversy.

Line 53: You say “reduction” twice in this sentence. “Dissimilatory iron reduction” is the reduction of Fe(III).

Line 55: You have discussed K⁺ earlier in the paper, therefore you don’t need to have this in parentheses here.

Line 55-56: This sentence needs to be referenced.

Line 57: What do you mean here by “ancient sediments”? i.e., you could say “sediments older than X billion years.” In any case, please remove “the” before “sediments.”

Line 86: You are missing the second hyphen after “(EDX).”

Line 92-93: As this is the results section, please remove your statement: “suggesting that organic carbon was oxidized...” This can be placed in your discussion section.

Line 102-104: Please remove this sentence, as it belongs in the discussion.

Line 105: Replace “difference” with “differences”.

Lines 105-110: Remove from your results, and place in discussion.

Lines 118-134: Your results of the different elements are not addressed in the discussion. Either include an analysis of these results in your discussion or remove this section from the main paper. This might be better placed in the supplemental section.

Line 143: Remove the word “significant,” unless you are referring to a statistical analysis.

Line 150: Replace “technique” with “techniques.”

Lines 207-211: You discussed the role of Fe(III) reduction in the introduction. I think these lines would be better suited in the intro rather than in the conclusions, unless you are making an additional point here?

Line 215-217: Please comment more on the timing of illitization processes. In order for clays to be transformed, K⁺ would need to come into physical contact with the smectite minerals in the surrounding sediments. However, if the microbes are adsorbing these cations into their cellular structure, then the cations would be sequestered inside the organisms and wouldn't be accessible by the clays. Is cell death therefore necessary for this process to be relevant to illitization? Alternatively, if K⁺ is sequestered on the surface of the cell membrane and/or trapped by the EPS, this could, in fact, imply that illitization occurs before cell death and before early diagenesis.

Lines 240-243: It may be necessary to think a bit more about the global implications of your research. Firstly, are there any calculations that can estimate the efficiency/magnitude of cation sequestration by microbes? Your argument would certainly be strengthened by providing your readers with some quantitative/semi-quantitative evaluation of the ability of microbial communities to sequester enough K^+ to affect the global reverse weathering cycle. I think one argument in your favor is simply the widespread nature of microbial mats on the seafloor during the Precambrian relative to post-Cambrian (so regardless of microbial efficiency in cation sequestration, the sheer abundance of these microorganisms could have facilitated a global change). This can also help to explain discrepancies between modern and ancient sediments.

Another thought is that if microbes are extremely efficient at sequestering K^+ from the seawater, it actually wouldn't take very high concentrations of this cation in the seawater/porewaters for the illitization process to commence. Thus, if the original concentrations of K^+ in the seawater were already low, and microbes were simply excellent scavengers of the low concentrations present, what role would these microbes play in global reverse weathering patterns?

Figure 3: In the figure itself, please replace "microbial mats" with "mat-related structures." Also, here you define the sediments as "host sediments" not "hosting sediments" as you do elsewhere in the manuscript.

Figure 4: It would be beneficial to include an XRF map for the sandstone in the main manuscript to show that both matrix samples (sandstone and black shale) are lower in potassium than corresponding MRS samples. Additionally, the scale for the black shale sample seems to be much larger (up to 900 μm) than for the pyritized MRS samples. Could this discrepancy account for the seemingly lower levels of potassium observed in the shale relative to the MRS? Furthermore, if you look at Supplementary Fig. 5a (the potassium subfigure), when observed on the same scale, concentrations of K^+ in the background shale and pyritized MRS seem to be quite similar. That is, of course, assuming that the background in this image is in fact the shale; it may not be if the MRS was extracted from its matrix material before analysis. Regardless, it may be beneficial to show all four XRF figures in the main manuscript, and at the same scale, so that readers can easily observe differences in K^+ across all sample types.

Reviewer #5 (Remarks to the Author):

This manuscript presents a comparison of Paleoproterozoic sediments with microbial biofilms and the corresponding strata that enclose the biofilms, from a 2.1 Ga age section in the west African Gabonese Republic. Based on clay mineral and chemical analyses aided by morphological observations, the authors argue that the Paleoproterozoic microbial mats extracted potassium from sea water and induced localized illitization during early low-temperature diagenesis. Illitization requires a reverse weathering process that the authors suggest is driven by active microbial uptake or passive sorption to microbial surfaces. The study compares smectite/illite abundance and crystallinity/crystal size between both pyritized and non-pyritized fossilized microbial mats and underlying sandstone and black shale. Relict microbial mat textures are associated with a.) large crystalline illite crystals, b.) enrichment in potassium, c.) smectite crystals indicating that potassium supply limited illitization. As the authors point out, it is well known that sandstones and shales derived from the same provenance would have different chemical composition. K, Na, Mg, and Ca highly soluble in near-surface, low-temperature environments; they are also soluble in low-temperature hydrothermal fluids, resulting in K-metasomatism, for example. Another point is that unless quartz is derived from massive quartz veins, there is no reservoir that can provide only quartz for dilution. This manuscript was initially reviewed by two specialists in the field, and both individuals agreed that the subject is of considerable interest to the mineral science community and is appropriate for publication in Nature Communications. However, both individuals also agreed that a number of corrections, modifications and clarifications are also needed, and described them in considerable detail. The authors have returned a revised manuscript and have provided accompanying text that addresses in considerable details how each suggestion and recommendation by the reviewers has been addressed. In my opinion the authors have done an excellent job in responding to the reviewers' comments and I would suggest that the manuscript is essentially acceptable for publication. However, I would like to offer a few, very minor comments that the authors may wish to consider as they prepare their final revision:

1. The authors might wish to reference "Clay minerals interaction with microorganisms: a review" by Cuadros (Clay Minerals, (2017) 52, 235–261)
2. Line 37 should read, "...rocks, such as komatiites....."
3. Line 105 should read, "The difference between....."
4. Lines 116-117 should read, "...comparisons show that the difference...."
5. Line 122 should read, "...also have a higher...."
6. Line 145 should read, "...maps reveal a higher K content...."
7. Line 149 should read, "...further analysed using the X-ray...."
8. Lines 152-153 should read, "...to characterize the illite content, with a higher R parameter corresponding to a higher illite content."
9. Line 197 should read, "...a limitation to no chemical alteration...."

Reviewer #1 (Remarks to the Author):

The authors provide a compelling example of microbial influence on early diagenesis, particularly related to clay diagenesis. This is a novel contribution that should promote interest and further study. The new data and revised text support a role for microbial contributions to reverse weathering processes. Importantly, the authors have addressed my concerns regarding K concentrations with the addition of synchrotron data. They did further diligence in addressing reviewer 2's (and reviewer 3) concerns about quartz dilution and I am satisfied with the support for their interpretation. The added information regarding the role of different microbial metabolisms, particularly as they relate to illitization, make these complex processes clear to the reader and more accessible to those scientists who are not as familiar with the particulars of environmental microbiology. This is a well supported and well written submission that I hope to see published due to its novel insight into microbe: mineral interactions in addition to its strong approach to interpreting those processes in ancient rock.

Response: We sincerely thank the reviewer for these positive remarks. Comments on this and previous version have helped to improve the content of the manuscript and provide better clarity.

Reviewer #3 (Remarks to the Author):

This is a much improved version of the original submission. Major comments* include the following: (1) Please pay attention to grammar/clarity issues.

(2) Please remove all discussion from your results section.

(3) There are several results that are not addressed in your discussion section, including your TEM data and elemental ratio plots. Please be sure to discuss the implications of all results.

(4) Your evidence for K⁺ uptake by microbial communities is convincing. However, the broader implications may require additional thought and discussion.

(5) Please address issues with XRF figures.

*All general comments are more thoroughly explained/detailed in the line-by-line comments (below).

Response: We thank the reviewer immensely for the depth of the comments provided on both rounds of review.

Line-by-line comments:

Line 1: The current paper does not describe clay formation, but rather the transformation from smectite to illite-rich clay layers. A reference to illitization or partial illitization (versus clay formation) would better describe this research.

Response: We proposed a new title as suggested by the associate editor (see lines 3-4). We think that the term "illitisation" is too specific. We agree that the illitisation is the transformation from smectite to illite-rich clay layers, but illite particles are neoformed during this process. Thus illitisation is a specific clay formation.

Line 24: Please remove the parentheses and incorporate the definition of illitization into the first sentence.

Response: We deleted the definition of illitisation process since the abstract must be less than 150 words. The definition and description of this mechanism is found in the first paragraph of the introduction (see lines 44-55).

Line 25: Replace “the smectite” with “a smectite.”

Response: We changed “the smectite” by “a smectite” (see line 26).

Line 26-27a: What do you mean here by “sediment pile”? You have not discussed sediment previously, only clay. Do you mean to say the clay-rich sediment on which the bacteria are living?

Response: We slightly revised the sentence (see lines 27-28). Here, we addressed the general problem of the study. Thus, we were referring more generally to sediments.

Line 26-27b: Also, is there a difference between “uptake” and “supply” to the sediment? If these are different process, please elaborate. If not, then please remove one of the two.

Response: We changed “uptake and supply to” by “uptake to” for clarity (see line 27).

Line 27c: You say “whether illitization was equally important in the geologic past.” Please clarify this statement. i.e., “equally important” to what?

Response: We deleted “equally” (see line 28). Regardless, this statement refers to the comparison with recent environments.

Line 28: Replace “2.1 billion-years” with “2.1-billion-year-old.”

Response: Changed to 2.1-billion-year-old (see line 29).

Line 31: Please elaborate here on what you mean by “in the MRS.” Do you mean physically associated with the filaments?

Response: We meant that the illite particles are abundant in the MRS lithology. We have no evidence that the particles are directly associated with microbial filaments. We added “facies” before “MRS” – “in the facies bearing MRS” (see line 33).

Line 32: I am unfamiliar with the term “hosting sediments.” Do you mean “host sediments?” If so, please replace here and elsewhere in the manuscript.

Response: Yes, we do. As suggested, we replaced “hosting sediments” with “host sediments” in the manuscript and figure captions.

Line 33: Replace “released it in the pore-waters” with “released it into the pore waters.”

Response: We changed “released it to the pore-waters” by “released it into the pore-waters” (see lines 35-36).

Lines 34-38: I find this sentence to be a bit confusing. Please re-word for clarity and check for grammar issues.

Response: The sentence has been modified (see lines 36-41).

Line 41: Remove "it" from this sentence.

Response: Deleted as suggested (see line 45).

Line 43-44: This entire sentence needs to be re-worded for clarity. Firstly, you mean to characterize the "illite-rich layers" in this sentence, rather than the "smectite-rich layers." By simply adding something to the effect of "the latter of which is characterized by..." this would clear up the first point of confusion as to what exactly you are characterizing. Secondly, if I am reading this sentence correctly, you mean to say that the illite-rich layers are composed of "interstratified I-S MLMs and long-range ordered I-S MLMs", i.e., the illite-rich layers are not comprised of 100% illite, and this only represents a partial illitization process. However, you add that these are actually "smectite-rich layers" as well; please clarify how layers can be both illite- and smectite-rich.

Response: We agree that the sentence was not clear. The illitisation process is characterized by the coexistence of smectite-rich, randomly interstratified I-S MLMs and long-range ordered I-S MLMs. This process is thought to be protracted with various MLMs co-existing in sediments. The trend towards increasing illite content is predicted by both solid-state transformation and dissolution-precipitation. Thus, we reworded the sentence for clarification (see lines 46-49).

Line 44-46: Add references regarding the "multiple mechanisms."

Response: References included (see line 51).

Line 46: Why are these proposed illitization mechanisms considered to be controversial? You say so here, but then only two give examples of mechanisms, and do not explain the controversy.

Response: Two main modes of transformation of hydrated smectite to illite have been proposed. There is no widely accepted mechanism because illitisation have been described in a wide range of environmental conditions. It appears that the mechanisms of conversion of smectite-rich to illite-rich layers could reflect specific environmental settings, but these are not clearly defined. Finally, "multiple" is inappropriate here. We reworded the sentence (see lines 49-51).

Line 53: You say "reduction" twice in this sentence. "Dissimilatory iron reduction" is the reduction of Fe(III).

Response: We modified the sentence (see line 59).

Line 55: You have discussed K⁺ earlier in the paper, therefore you don't need to have this in parentheses here.

Response: Corrected (see line 63).

Line 55-56: This sentence needs to be referenced.

Response: A reference, which has been proposed by the reviewer#5, is in line with the sentence (see line 65). The review by Cuadros (2017) discussed the microbe-clay interaction in depth and the implication on weathering.

Cuadros, J. Clay minerals interaction with microorganisms: a review. Clay Miner. 52, 235–261 (2017).

Line 57: What do you mean here by “ancient sediments”? i.e., you could say “sediments older than X billion years.” In any case, please remove “the” before “sediments.”

Response: Here, ancient sediments referred to sediments that have undergone diagenesis (see lines 65-67).

Line 86: You are missing the second hyphen after “(EDX).”

Response: It is not needed here since the first hyphen is used for spelling the abbreviation SEM-EDS (see line 98).

Line 92-93: As this is the results section, please remove your statement: “suggesting that organic carbon was oxidized...” This can be placed in your discussion section.

Response: Moved as suggested to discussion (see lines 204-207).

Line 102-104: Please remove this sentence, as it belongs in the discussion.

Response: Lines moved to discussion (see lines 207-210).

Line 105: Replace “difference” with “differences”.

Response: Corrected (see line 210).

Lines 105-110: Remove from your results, and place in discussion.

Response: Lines moved to discussion (see lines 210-215).

Lines 118-134: Your results of the different elements are not addressed in the discussion. Either include an analysis of these results in your discussion or remove this section from the main paper. This might be better placed in the supplemental section.

Response: Following this comment, the geochemical comparison of selected major elements has been placed in the supplementary information (see lines 133-150, and Supplementary Note 1).

Line 143: Remove the word “significant,” unless you are referring to a statistical analysis.

Response: Removed as advised and revised to: “are not enriched in K” (see lines 158-159).

Line 150: Replace “technique” with “techniques.”

Response: As suggested by the reviewer #5, we should have stated “the X-ray diffraction (XRD) technique” (see line 166).

Lines 207-211: You discussed the role of Fe(III) reduction in the introduction. I think these lines would be better suited in the intro rather than in the conclusions, unless you are making an additional point here?

Response: The reviewer is correct that we already discussed the involvement of iron-reducing microbes in the release of structural Fe(III). Thus, the sentence has been moved into the introduction and slightly revised (see lines 60-62).

Line 215-217: Please comment more on the timing of illitization processes. In order for clays to be transformed, K⁺ would need to come into physical contact with the smectite minerals in the surrounding sediments. However, if the microbes are adsorbing these cations into their cellular structure, then the cations would be sequestered inside the organisms and wouldn't be accessible by the clays. Is cell death therefore necessary for this process to be relevant to illitization? Alternatively, if K⁺ is sequestered on the surface of the cell membrane and/or trapped by the EPS, this could, in fact, imply that illitization occurs before cell death and before early diagenesis.

Response: This is good point, but we do not have any relevant data to resolve between illite formed when cells were still alive vs. after their death. Illitisation is a protracted process, taking place throughout diagenesis (see lines 44-46, and references cited). The proportion of illite in the MLMs increases as a function of temperature, pressure, time, and other chemical variables. This is a depth-dependant process, occurring in a response to burial. Our model involves biologically derived K⁺ and its release to the pore-waters during both decay of microbial mat organic matter and early diagenesis. Regardless of the path for K⁺ sequestration, the illitisation cannot take place before diagenesis. In addition, the micrometric-sized illite particles in the MRS facies further support the tendency of dissolution-precipitation processes following the Ostwald ripening-like rule.

Lines 240-243: It may be necessary to think a bit more about the global implications of your research. Firstly, are there any calculations that can estimate the efficiency/magnitude of cation sequestration by microbes? Your argument would certainly be strengthened by providing your readers with some quantitative/semi-quantitative evaluation of the ability of microbial communities to sequester enough K⁺ to affect the global reverse weathering cycle.

Response: The reviewer is absolutely correct that the comparison of intracellular concentration of K⁺ and seawater composition would help the readers to evaluate the ability of microbes to accumulate K⁺. Accordingly, we have provided pertinent information on K content in cell cytoplasm and seawater (see lines 254-254).

The K content in modern seawater is 10 times lower than in microbial cells (100 mM in cell cytoplasm; Mulkidjanian et al., 2012). As microbes efficiently sequester K⁺ from modern seawater, one may assume that they significantly contributed to reverse weathering by enhancing K supply to sediments during diagenesis.

Mulkidjanian, A. Y., Bychkov, A. Y., Dibrova, D. V., Galperin, M. Y. & Koonin, E. V. Origin of first cells at terrestrial, anoxic geothermal fields. Proc. Natl. Acad. Sci. 109, E821–E830 (2012).

I think one argument in your favor is simply the widespread nature of microbial mats on the seafloor during the Precambrian relative to post-Cambrian (so regardless of microbial efficiency in cation sequestration, the sheer abundance of these microorganisms could have facilitated a global change). This can also help to explain discrepancies between modern and ancient sediments.

Response: We thank the reviewer for bringing this point to our attention, and we now incorporated into the Discussion (see lines 287-291).

Another thought is that if microbes are extremely efficient at sequestering K⁺ from the seawater, it actually wouldn't take very high concentrations of this cation in the seawater/porewaters for the illitization process to commence. Thus, if the original concentrations of K⁺ in the seawater were already low, and microbes were simply excellent scavengers of the low concentrations present, what role would these microbes play in global reverse weathering patterns?

Response: The K content of Precambrian seawater is uncertain although it is logical to assume larger content of K in Precambrian seawater since the continental margin repository for marine evaporites was smaller (see lines 283-287, and references cited).

Figure 3: In the figure itself, please replace “microbial mats” with “mat-related structures.” Also, here you define the sediments as “host sediments” not “hosting sediments” as you do elsewhere in the manuscript.

Response: We replaced “microbial mats” with “MRS” and we left “host sediments” in the figure itself as we now changed ‘hosting sediments’ to ‘host sediments’ in the text.

Figure 4: It would be beneficial to include an XRF map for the sandstone in the main manuscript to show that both matrix samples (sandstone and black shale) are lower in potassium than corresponding MRS samples.

Response: We are unable to provide XRF elemental maps for sandstone because the non-pyritized MRS samples have been removed from their host sediments during sample preparation for XRF analyses. Besides, the XRF measurements were performed before the first submission, and, at that time, we did not analyse the K content in sandstone with this analytical technique.

Additionally, the scale for the black shale sample seems to be much larger (up to 900 μm) than for the pyritized MRS samples. Could this discrepancy account for the seemingly lower levels of potassium observed in the shale relative to the MRS?

Indeed, the measured area of the black shale sample is larger. XFM is a scanning imaging method, where the image is measured pixel by pixel. As a result, the image size has no effect on the intensity value of a pixel; it is the total exposure time necessary to measure a larger sample area, which is

proportionally longer. In our study each elemental map was normalized to 10 ms/pixel exposure time, independently of the total measurement time, allowing for direct comparison of the different samples. Following the suggestion of the referee, a common intensity scale was chosen for the K distribution maps for easier comparison (Figure 4).

Furthermore, if you look at Supplementary Fig. 5a (the potassium subfigure), when observed on the same scale, concentrations of K⁺ in the background shale and pyritized MRS seem to be quite similar. That is, of course, assuming that the background in this image is in fact the shale; it may not be if the MRS was extracted from its matrix material before analysis.

Response: The reviewer is correct that the background in this image displays a higher K content, potentially reflecting contamination with the MRS as mentioned on lines 130-132. Regardless, the biologically derived K was released to the surrounding sediments, and it is not surprising that the particles few μm below the pyritized mat structure have been transformed into illite.

Regardless, it may be beneficial to show all four XRF figures in the main manuscript, and at the same scale, so that readers can easily observe differences in K⁺ across all sample types.

Response: We agree and show the S and K elemental maps with the larger scale in the main text of the revised version as Figure 4. The images have now the same common intensity scale for the K distribution maps. We removed the small-sized elemental maps as these are not critical.

Reviewer #5 (Remarks to the Author)

This manuscript presents a comparison of Paleoproterozoic sediments with microbial biofilms and the corresponding strata that enclose the biofilms, from a 2.1 Ga age section in the west African Gabonese Republic. Based on clay mineral and chemical analyses aided by morphological observations, the authors argue that the Paleoproterozoic microbial mats extracted potassium from sea water and induced localized illitization during early low-temperature diagenesis. Illitization requires a reverse weathering process that the authors suggest is driven by active microbial uptake or passive sorption to microbial surfaces. The study compares smectite/illite abundance and crystallinity/crystal size between both pyritized and non-pyritized fossilized microbial mats and underlying sandstone and black shale. Relict microbial mat textures are associated with a.) large crystalline illite crystals, b.) enrichment in potassium, c.) smectite crystals indicating that potassium supply limited illitization. As the authors point out, it is well known that sandstones and shales derived from the same provenance would have different chemical composition. K, Na, Mg, and Ca highly soluble in near-surface, low-temperature environments; they are also soluble in low-temperature hydrothermal fluids, resulting in K-metasomatism, for example. Another point is that unless quartz is derived from massive quartz veins, there is no reservoir that can provide only quartz for dilution. This manuscript was initially reviewed by two specialists in the field, and both individuals agreed that the subject is of considerable interest to the mineral science community and is appropriate for publication in Nature Communications. However, both individuals also agreed that a number of corrections, modifications and clarifications

are also needed, and described them in considerable detail. The authors have returned a revised manuscript and have provided accompanying text that addresses in considerable details how each suggestion and recommendation by the reviewers has been addressed. In my opinion the authors have done an excellent job in responding to the reviewers' comments and I would suggest that the manuscript is essentially acceptable for publication.

Response: We sincerely acknowledge the reviewer for the positive comments.

However, I would like to offer a few, very minor comments that the authors may wish to consider as they prepare their final revision:

1. The authors might wish to reference "Clay minerals interaction with microorganisms: a review" by Cuadros (Clay Minerals, (2017) 52, 235–261).

Response: The reviewer is absolutely correct for proposing this reference. This reference provides the scientific background of the microbe-clay interactions, including the microbially mediated dissolution and precipitation of phyllosilicates. Moreover, the author of this publication has inferred that "if K or NH_4 are available during the Fe-reduction process in smectite, the formation of illite will follow", which is perfectly in line with our study. Cuadros (2017) even discussed the indirect contribution of bacterial activity to the reverse weathering, but without alluding to biological feedback on the atmospheric composition and seawater chemistry.

2. Line 37 should read, "...rocks, such as komatiites...."

Response: We reworded the sentence as suggested by the reviewer #3 (see lines 36-41).

3. Line 105 should read, "The difference between...."

Response: Corrected (see line 210). The sentence has been moved to the discussion, as suggested by the reviewer #3.

4. Lines 116-117 should read, "...comparisons show that the difference...."

Response: Corrected (see line 129).

5. Line 122 should read, "...also have a higher...."

Response: Corrected. The paragraph has been moved to the Supplementary Note 1, as suggested by the reviewer #3.

6. Line 145 should read, "...maps reveal a higher K content...."

Response: Corrected (see line 161).

7. Line 149 should read, "...further analysed using the X-ray...."

Response: Corrected (see line 166).

8. Lines 152-153 should read, "...to characterize the illite content, with a higher R parameter corresponding to a higher illite content."

Response: Corrected (see line 169).

9. Line 197 should read, "...a limitation to no chemical alteration..."

Response: Corrected (see line 225).